# Microbiome and metabolome features in inflammatory bowel disease via multi-omics integration analyses across cohorts

Lijun Ning[1,6], Yi-Lu Zhou[1,6], Han Sun[2,6], Youwei Zhang[3,6], Chaoqin Shen[4,6], Zhenhua Wang[1], Baoqin Xuan[1], Ying Zhao[1], Yanru Ma[1], Yuqing Yan[1], Tianying Tong[1], Xiaowen Huang[1], Muni Hu[1], Xiaoqiang Zhu[1], Jinmei Ding[1], Yue Zhang[1], Zhe Cui[5], Jing-Yuan Fang ⑩[1], Haoyan Chen ⑩[1]✉ & Jie Hong ⑩[1]✉

The perturbations of the gut microbiota and metabolites are closely associated with the progression of inflammatory bowel disease (IBD). However, inconsistent findings across studies impede a comprehensive understanding of their roles in IBD and their potential as reliable diagnostic biomarkers. To address this challenge, here we comprehensively analyze 9 metagenomic and 4 metabolomics cohorts of IBD from different populations. Through cross-cohort integrative analysis (CCIA), we identify a consistent characteristic of commensal gut microbiota. Especially, three bacteria, namely *Asaccharobacter celatus*, *Gemmiger formicilis*, and *Erysipelatoclostridium ramosum*, which are rarely reported in IBD. Metagenomic functional analysis reveals that essential gene of Two-component system pathway, linked to fecal calprotectin, are implicated in IBD. Metabolomics analysis shows 36 identified metabolites with significant differences, while the roles of these metabolites in IBD are still unknown. To further elucidate the relationship between gut microbiota and metabolites, we construct multi-omics biological correlation (MOBC) maps, which highlights gut microbial biotransformation deficiencies and significant alterations in aminoacyl-tRNA synthetases. Finally, we identify multi-omics biomarkers for IBD diagnosis, validated across multiple global cohorts (AUROC values ranging from 0.92 to 0.98). Our results offer valuable insights and a significant resource for developing mechanistic hypotheses on host-microbiome interactions in IBD.

Recent studies have revealed that alterations in gut microbiota and metabolites are linked to changes in human health and various diseases, including Inflammatory bowel disease (IBD)[1,2]. IBD is a chronic inflammatory condition that affects the gastrointestinal tract and includes two main forms: Crohn's disease (CD) and ulcerative colitis (UC)[3,4]. Millions of people worldwide are affected by IBD, and its incidence is shifting from developed to developing countries, highlighting the importance of early diagnosis[5–7]. The utilization of fecal shotgun metagenomics provides a powerful means to identify disease-associated species and understand co-metabolism between the host and microbiota at a higher taxonomic resolution[8], while metabolomics reveals changes in gut metabolites as a messenger of information exchange between the gut microbiota and the host[9]. Combining metagenomics and metabolomics presents a promising approach for understanding the development of IBD and related gut environment alterations and offers a non-invasive biomarker for IBD.

Previous research has reported some signatures in gut microbiota and metabolites[10–12], but the differences among various studies have made it challenging to validate these signatures across diverse groups, reducing the diagnostic value of the microbiome and metabolome in IBD. Therefore, there is an urgent need for multi-national, large-scale cohorts, multiomics characterization, standardized sampling and analysis, as well as model systems to uncover the relationship between gut microbiota and their functions with gut metabolites[13]. Cross-cohort integrative analysis (CCIA) is expected to address these challenges by assessing the robustness of disease-microbiome associations through the comparison of several case-control studies. The goal of CCIA is to identify consistent associations across various cohorts, thus minimizing the impact of biological or technical confounders. The remarkable performance of CCIA in some prior studies highlights its potential as a valuable tool in various research fields[14–16].

In this work, we comprehensively analyzed nine metagenomic cohorts ($N = 1363$ cases) and four metabolomics cohorts ($N = 398$ cases) of IBD patients from different countries or regions through CCIA. Our objective was to identify specific gut bacteria, metabolites, and their associated Kyoto Encyclopedia of Genes and Genomes (KEGG) orthology (KO) genes that contribute to the development of IBD across diverse cohorts. We also aimed to create diagnostic models using disease-specific biomarkers from diverse cohorts to enhance IBD diagnosis. Furthermore, we sought to clarify the intricate relationships among these bacteria, metabolites, and KO genes within the context of IBD. Ultimately, our research holds the potential to provide fresh insights for the diagnosis and treatment of IBD.

## Results

### Workflow for cross-cohort integration analysis of fecal metagenomics and metabolomics in IBD

In this study, we employed a multi-omics approach that integrates fecal metagenomics and metabolomics to investigate alterations in the gut microbiota of IBD. A total of 9 metagenomic cohorts from four different regions or countries ($n = 1363$ cases) were included in this study. These cohorts were divided into six discovery cohorts and three validation cohorts (Fig. 1a, Supplementary Fig. 1b and Supplementary Table 1). Additionally, we included four metabolomic cohorts ($n = 398$ cases), of which two external cohorts were examined using non-targeted metabolomics, and two in-house cohorts were examined using targeted metabolomics (Fig. 1a).

To ensure consistency in the bioinformatic analyses, we applied the MetaPhlan3 tool for taxonomic profiling and HUMAnN3 for functional profiling to reprocess all raw sequencing data. Furthermore, by annotating metabolite names with a unified ID using the Human Metabolome Database (HMDB), we identified 79 metabolites that were shared among the four cohorts. These metabolites will be utilized in a cross-cohort analysis of metabolomics data (Fig. 1b).

Furthermore, our aim is to reveal the patterns of variation in gut microbiota and fecal metabolites through a comprehensive statistical analysis, and then utilize machine learning techniques for diagnosing IBD. Initially, we excluded samples that were repeatedly collected from external cohorts. Then, we utilized a sequence of differential analyses and feature selection to identify 31 species, 25 functional genes, and 13 metabolites that were effective in diagnosing IBD patients. Subsequently, we selected four cohorts containing both metagenomic and metabolomic data ($n = 391$ cases) for integrated analysis, enabling us to establish the most precise diagnostic model. Finally, to explore gut microbiota-related metabolic processes, we introduced the multi-omics biological correlation (MOBC) maps framework (Fig. 1c).

### Identification of bacterial biomarkers at the species level for diagnosing IBD through cross-cohorts

The main goal of the CCIA was to discover particular gut microbial species that exhibited consistent alterations in abundance in

metagenomes of individuals with IBD. Prior to the analysis, it was imperative to evaluate the impact of cohort-related variations (CRV) on the microbiome composition since there were differences among the cohorts in terms of biological and technical factors. To mitigate the impact of confounding variables, we compared the identified gut microbial species with other factors such as patient age, gender, cohort, country, body mass index (BMI), and antibiotic use. Our analysis revealed that the factor of " cohort" or "country" had a significant impact on the species composition (Supplementary Fig. 2a, b). Therefore, to minimize potential biases, we restricted subsequent analysis to patients within the same country in a single cohort and processed the metagenomic data using the same analysis method.

We next evaluated the alpha diversity by measuring the Shannon and Simpson index, and found that IBD patients exhibited lower microbial alpha diversity compared to healthy controls (FDR < 0.0001) (Fig. 2b, c). In addition, it's important to note that the dissimilarities in beta diversity, which were calculated using the Bray-Curtis distance metric, were found to vary not only based on the disease status (as indicated by a PERMANOVA analysis with $P = 0.001$) but also across different cohorts (as indicated by a PERMANOVA analysis with $P = 0.001$) (Fig. 2d). Thus, reduced microbial diversity observed in IBD patients may play a significant role in disrupting the delicate balance of the gut ecosystem and could be a contributing factor to the development of IBD.

To identify potential microbial biomarkers for the diagnosis of IBD, we employed a method provided by a previous study to analyze the composition of microbial species[15]. Through CCIA with an FDR less than 0.0001, 74 microbial species were identified to have significantly different abundances in the gut microbiomes (Fig. 2e and Supplementary Table 2). Despite significant differences in diet and genetics among IBD patients from various regions or countries, a consistent pattern of alteration in their gut microbiota was observed. Our results suggest that IBD patients exhibit a significant reduction in commensal gut microbiota, which are crucial for various activities of their hosts. Several species, known to produce butyric acid[17,18], were found to be depleted in the gut microbiota of IBD, including *Faecalibacterium prausnitzii*, *Roseburia intestinalis*, *Eubacterium hallii*, *Gemmiger formicilis*, *Eubacterium rectale*, and *Ruminococcus bromii* (Fig. 2e). In addition, our study suggests that certain bacteria involved in other intestinal metabolism, including those associated with mineral metabolism (such as *Collinsella aerofaciens* involved in iron metabolism[19]), bile acid metabolism[20] (such as *Ruminococcus torques*), and urea cycle metabolism[21] (such as *Bifidobacterium longum*), all of which demonstrate a significant decrease (Fig. 2e). Some bacteria that antagonize pro-inflammatory microorganisms are also significantly reduced. *Alistipes putredinis* showed a negative correlation with the colonization of *Candida albicans*, which is enriched in the gut of IBD patients and exacerbates intestinal inflammation by inducing Th17 cell differentiation[22,23]. Notably, through a comprehensive analysis, we have successfully identified two specific microbial species, *Asaccharobacter celatus* and *Gemmiger formicilis*, which are depleted across six distinct IBD cohorts (Supplementary Fig. 2c). However, previous studies have not extensively investigated or emphasized the relationship between these bacteria and IBD. *Asaccharobacter celatus* possesses the ability to produce Equol[24], which has the potential to alleviate experimental autoimmune encephalomyelitis in mice[25]. This implies a potential role for *Asaccharobacter celatus* in the regulation of autoimmune diseases.

Furthermore, our study also revealed that certain bacteria, including *Ruminococcus gnavus*, *Bacteroides fragilis*, *Escherichia coli*, and *Clostridium innocuum* were consistently enriched in IBD (Fig. 2e). *Ruminococcus gnavus* can produce pro-inflammatory polysaccharides and mucin-degrading trans-sialidase, which disrupts the intestinal mucosal barrier and promotes inflammation[26–28]. In clinical practice, special attention should be paid to *Clostridium innocuum* infection in

## (a) Participant recruitment and Data collection

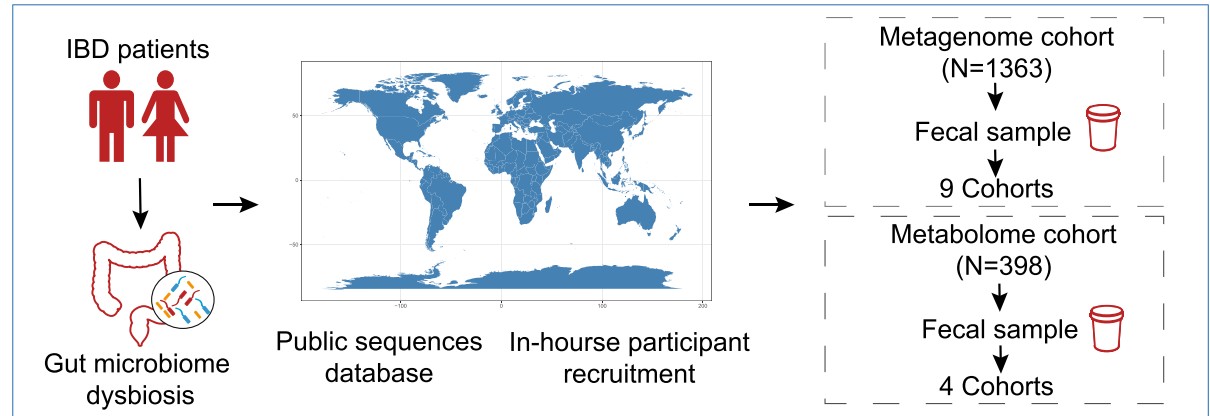

## (b) Cumputational pipeline

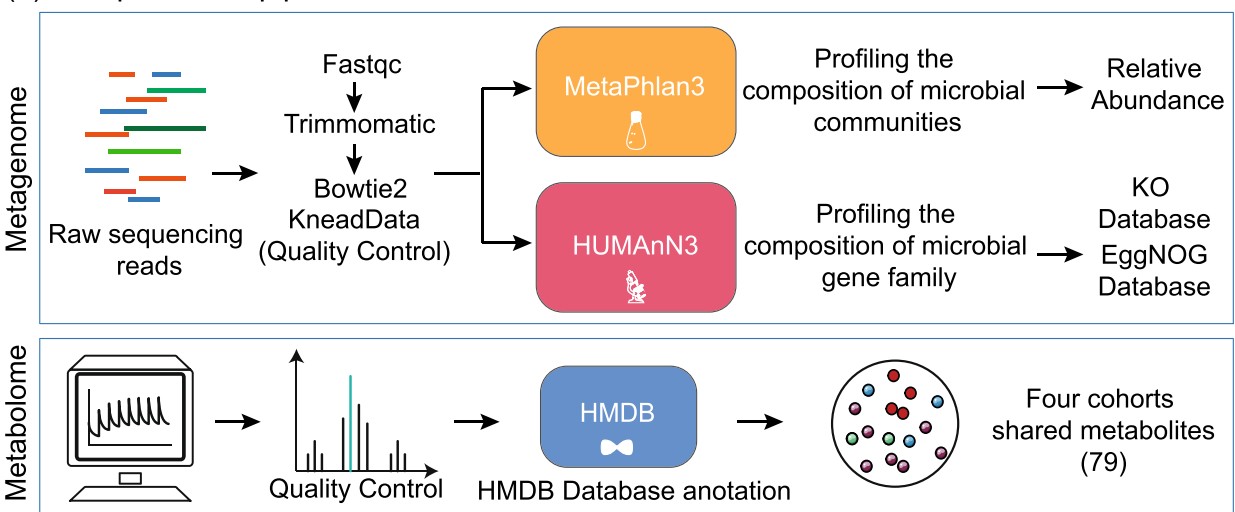

## (c) Data Analysis

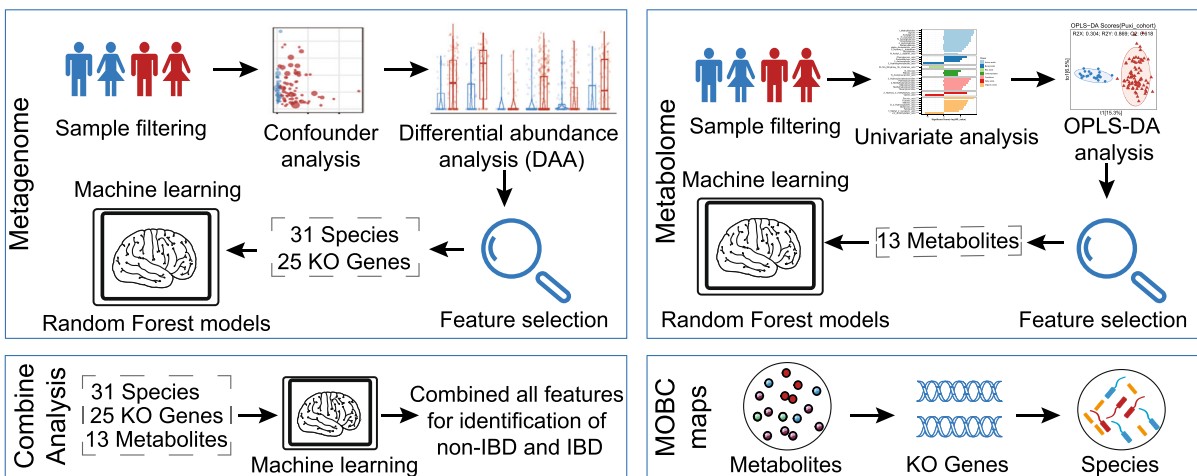

**Fig. 1 | Workflow for cross-cohort integration analysis of fecal metagenomics and metabolomics in IBD. a** We included a total of 9 fecal metagenomic cohorts ($n = 1363$) and 4 metabolomic cohorts ($n = 398$) from diverse geographic locations worldwide. **b** We utilized the MetaPhlan3 tool for taxonomic profiling and HUMAnN3 for functional profiling to reprocess all raw metagenomic sequencing data. Additionally, we annotated the compound names from the metabolomics analysis with the same ID number using the HMDB (Human Metabolome Database). **c** Through strict sample filtering, detailed bioinformatics analysis, and feature selection, we identified a series of representative features, including 31 bacterial features, 25 KO genes, and 13 metabolites. Subsequently, we developed machine learning models and used the features for the diagnosis of IBD. Finally, we introduced the multi-omics biological correlation (MOBC) maps framework to shed light on the interconnected relationships among gut bacteria, metabolites, and KO genes in IBD.

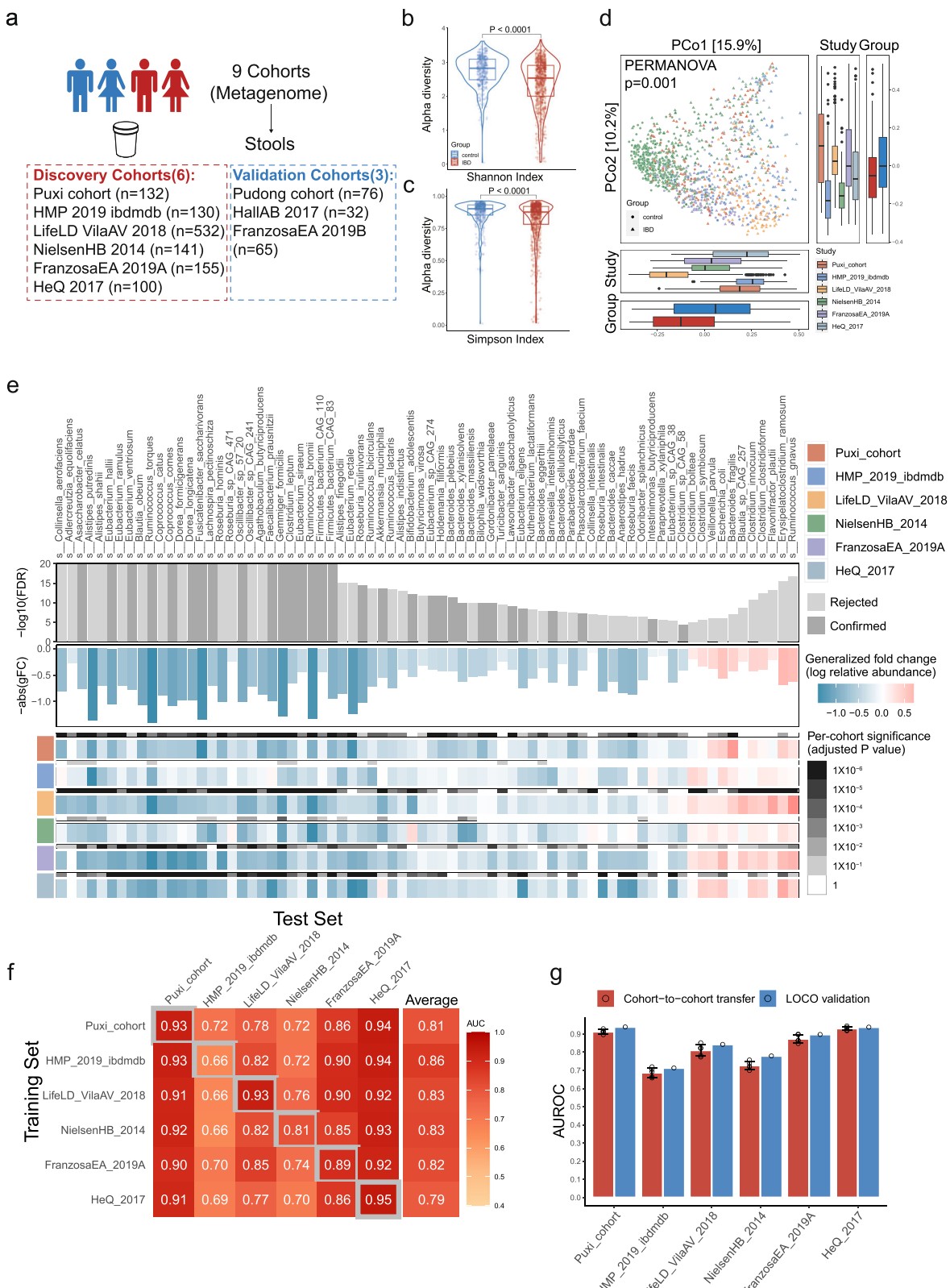

IBD, as it has been associated with creeping fat and intestinal strictures in Crohn's disease and is inherently resistant to vancomycin[29]. In addition, *Erysipelatoclostridium ramosum*, a bacterium that has been documented sparsely in the studies, but appears to be more prevalent in IBD, as confirmed in multiple cohorts (Supplementary Fig. 2d). The role of this bacterium in the pathogenesis and progression of IBD is not yet fully understood, and therefore requires further investigation.

Overall, our findings highlight the significant alterations in the gut microbiota of patients with IBD and the potential role of specific bacterial groups in the pathogenesis of this disease.

Subsequently, we utilized a machine learning method (Random Forest, RF), for the diagnosis of IBD. To enhance the accuracy and interpretability of our model, as well as minimize the impact of redundant and irrelevant features, we utilized the Iterative Feature

**Fig. 2 | Identification of bacterial biomarkers at the species level for diagnosing IBD through cross-cohorts. a** Overall composition of the population across 9 metagenomic datasets ($n = 1363$). **b**, **c** The alpha diversity of IBD (red, $n = 795$) and control (blue, $n = 395$) was measured using the Shannon index and Simpson index. The adjusted $p$ value (two-sided test) was calculated using MMUPHin tools. The data in boxplots is represented using interquartile ranges (IQRs), with the median shown as a horizontal line, and the whiskers extending to the most extreme points within 1.5 times the IQR. Exact $p$ values are provided in the Source data file. **d** Principal coordinate analysis (PCoA) shows significant differences in microbial composition between both groups ($P = 0.001$) and cohorts ($P = 0.001$). The significance of beta diversity based on Bray-Curtis distance was calculated using PERMANOVA with 999 permutations (two-sided test, $n = 1190$). The data in boxplots is represented using interquartile ranges (IQRs), with the median shown as a black horizontal line, and the whiskers extending to the most extreme points within 1.5 times the IQR. **e** The top bar graph displays the 74 gut bacterial species with the most significant differences ($P < 0.0001$), as calculated using a two-sided Wilcoxon

test with FDR-corrected P values. Among these species, 31 are highlighted in dark gray as feature species for subsequent random forest modeling (Confirmed). The middle bar graph shows the generalized fold change (gFC) of these 74 significant species, with red indicating 11 species that are enriched in IBD and blue indicating 63 species that are depleted in IBD. At the bottom, heatmaps are shown in gray and in color, respectively, displaying the species-level significance and the gFC within individual cohorts. (**f**) The classification models accuracy of IBD resulting from 10-fold cross-validation was assessed within each cohort (gray boxes along the diagonal), as well as cohort-to-cohort model transfer (external validations off-diagonal), using the AUROC for classifiers trained on species abundance profiles. **g** The classification models accuracy, as evaluated by AUROC on a hold-out cohort, improves when taxonomic data from all other cohorts are combined for training using leave-one-cohort-out (LOCO) validation, compared to models trained on data from a single cohort (cohort-to-cohort transfer). The error bars indicate the mean ± sd, $n = 5$. Source data are provided as a Source Data file.

Elimination (IFE) technique to perform feature selection. As a result, we rigorously selected 31 feature species from the 74 differentially abundant species (Fig. 2e), mainly from the phylum Firmicutes, for modeling analysis (Supplementary Fig. 2e). We initially developed a random forest model using 10-fold cross-validation with 31 signature species from the 6 cohorts, which demonstrated strong ability to detect IBD across all cohorts, with AUROC ranging from 0.66 to 0.95 (Fig. 2f). However, upon analyzing the performance of the classifiers on the six cohorts, we found that the IBDMDB cohort exhibited significantly worse results than the other five cohorts. This variation in results may be due to the fact that the patients were from different medical centers, despite being from the same country. Such variability and heterogeneity in the data may have contributed to the reduced accuracy of the classifiers. Moreover, To evaluate the transferability and geographical diversity of the identified features for diagnosing IBD, we performed cohort-to-cohort transfer analysis and leave-one-cohort-out (LOCO) analysis using established methods[30]. The species-level models demonstrated an average cohort-to-cohort transfer analysis performance ranging from 0.79 to 0.86 in terms of AUROC, with the majority of values hovering around 0.8 (Fig. 2f). Our analysis demonstrated that the LOCO analysis performance ranged from 0.71 to 0.94 (Fig. 2g).

In addition, to validate the accuracy and transferability of our model in independent cohorts, we included three independent IBD metagenomic cohorts (Fig. 2a). Specifically, the HallAB 2017 cohort had an average AUROC of 0.70, the FranzosaEA 2019B cohort had an average AUROC of 0.90, and the Pudong cohort had an average AUROC of 0.89 (Supplementary Fig. 2f). However, the AUROCs for the LOCO analysis slightly improved, with HallAB 2017 at 0.72, FranzosaEA 2019B at 0.96, and the Pudong cohort at 0.91 (Supplementary Fig. 2f). If we consider the HallAB 2017 cohort as an outlier, the independent validation of our models can result in an AUROC of approximately 0.90.

Since previous research has revealed that changes in the microbiome can be associated with various diseases, emphasizing the importance of identifying disease-specific microbiome signatures. We next investigated the false positive rate (FPR) of our metagenomic classifiers by analyzing metagenomes from patients with gastrointestinal (GI) diseases, such as adenoma and colorectal cancer (CRC), as well as non-GI diseases like type 2 diabetes (T2D). Therefore, we utilized LOCO classification models of species, calibrated to achieve a FPR of 0.08 and 0.13 on CRC datasets (Supplementary Fig. 2g), respectively. We found that the FPRs on the other disease datasets were also relatively low, with adenoma at 0.15 and T2D at 0.11 (Supplementary Fig. 2g). These results suggest that our models have excellent disease specificity. Overall, our findings demonstrate that our model has excellent specificity and can accurately identify disease-specific microbiome signatures.

## Identification of IBD diagnostic markers by demonstrating variations of Kyoto Encyclopedia of Genes and Genomes (KEGG) orthology (KO) across different IBD cohorts

Metagenomic functional analysis plays a critical role in understanding the complex interactions between the gut microbiome and human health. In our study, we first annotated gene families obtained from metagenomic analysis as KO genes using the KEGG orthology database, resulting in 9,270 KO genes. We applied a low-abundance filtering step to obtain a final set of 3,732 KO genes, followed by differential abundance analysis using the same method as earlier. To avoid overfitting of the model, we used a strict FDR approach (CCIA analysis FDR $< 1 \times 10^{-12}$), which led to the identification of 162 differentially expressed KO genes between normal and IBD patients (Fig. 3a and Supplementary Tables 3). Most of these KO genes are classified as potentially coding for metabolic enzymes, particularly those involved in amino acid and other metabolism. This indicates a strong association between gut microbiota and their metabolic activities.

Subsequently, an enrichment analysis was conducted on these genes, uncovering 12 pathways with potential implications for both gut microbiota and disease (FDR $< 0.05$). Among these pathways, four of them displayed an upregulated pattern, including "Cell cycle – Caulobacter," "Two-component system," "Lipopolysaccharide biosynthesis," and "Aminoacyl-tRNA biosynthesis" (Supplementary Fig. 3a). Specifically, Two-component systems play a critical role in regulating virulence factors in certain bacteria[31–33]. Among the 8 KO genes enriched in the Two-component system pathway, *crp* (K10914, CRP/FNR family transcriptional regulator) stands out as the gene with the highest generalized fold change (Supplementary Fig. 3b). Previous studies have indicated its potential involvement in various biological processes, such as osmoregulation[34], stringent response[35], and biofilm formation[36]. Furthermore, the relative abundance of *crp* is positively correlated with fecal calprotectin (Supplementary Fig. 3c). These findings suggest that *crp* could potentially act as a critical transcriptional regulatory factor contributing to the occurrence of gut inflammation. Additionally, we have also observed the downregulation of several pathways in IBD, such as Propanoate metabolism, Phosphotransferase system (PTS), Styrene degradation, Glycolysis / Gluconeogenesis, and Biosynthesis of amino acids (Supplementary Fig. 3a). It has been reported that Akkermansia-related propanoate metabolism enhanced the development of the intestinal epithelium through intestinal stem cell-mediated mechanisms, where intestinal stem cells (ISCs) play a vital role in the developmental processes and swift regeneration of the intestinal lining[37].

Additionally, we sought to assess the potential diagnostic usefulness of the KO genes in IBD. We utilized the IFE method for feature selection and identified 25 KO genes as features for random forest modeling out of the 162 KO genes. After conducting a 10-fold cross-validation analysis, we observed that all cohorts, except for the HeQ

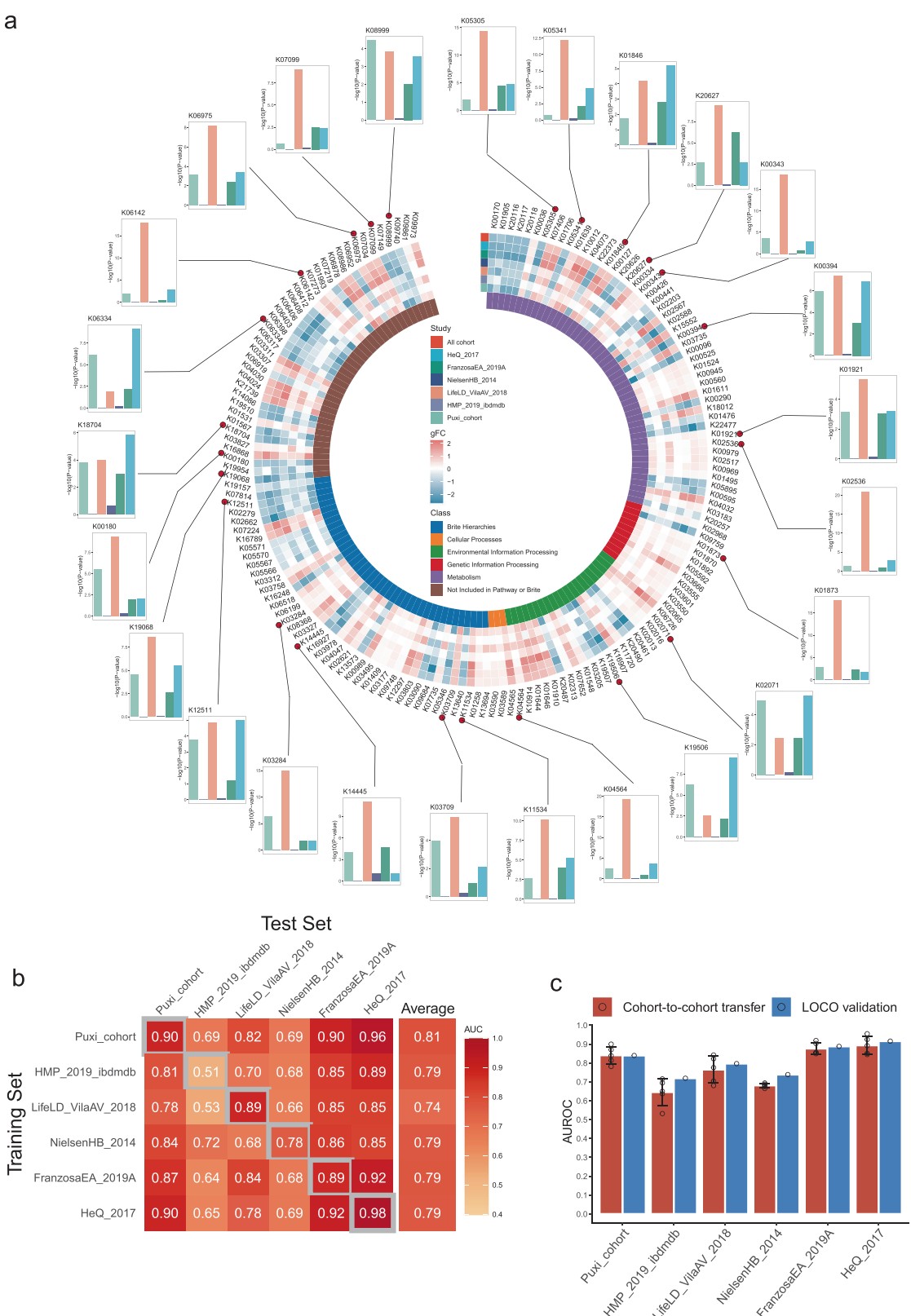

2017 cohort that displayed excellent diagnostic performance (AUROC: 0.98), showed a decrease in AUROC values compared to bacterial species models. The cohort-to-cohort transfer analysis demonstrated that the mean AUROC values for all cohorts ranged from 0.74 to 0.81 (Fig. 3b). Consistent with bacterial species models, the LOCO analysis showed that the diagnostic value of KO genes was slightly higher than in the cohort-to-cohort transfer analysis (Fig. 3c). In general, while the

diagnostic performance of KO gene models is somewhat lower than that of bacterial species models, their diagnostic sensitivity is still acceptable.

Furthermore, to validate the diagnostic potential of the 25 KO genes, we applied them to the above independent cohorts. In cohort-to-cohort transfer models, the AUROCs ranged from 0.71 to 0.96, while in LOCO classification models, the AUROCs ranged from 0.61 to 0.87

**Fig. 3 | Identification of IBD diagnostic markers by demonstrating variations of Kyoto Encyclopedia of Genes and Genomes (KEGG) orthology (KO) across different IBD cohorts. a** The circular complex heatmap displays the 162 most significant KO genes ($P < 1×10^{-12}$), as calculated using a two-sided Wilcoxon test with FDR-corrected P values in the cross-cohort analysis. The inner-circle heatmap shows the generalized fold change (gFC) values of these 162 KO genes, with red indicating enriched and blue indicating depleted in IBD. The outer-circle bar chart displays the p-values of 25 featured KO genes in each cohort used for modeling. **b** The IBD classification accuracy was measured using AUROC for classifiers trained

on the KO genes abundance profiles. The classification accuracy was evaluated using 10-fold cross-validation within each cohort (gray boxes along the diagonal) and cohort-to-cohort model transfer (external validations off-diagonal). **c** The classification models accuracy, as evaluated by AUROC on a hold-out cohort, improves when functional data (KO genes) from all other cohorts are combined for training using leave-one-cohort-out (LOCO) validation, compared to models trained on data from a single cohort (cohort-to-cohort transfer). The error bars indicate the mean ± sd, $n = 5$. Source data are provided as a Source Data file.

(Supplementary Fig. 3d). We also tested for FPR by analyzing other disease datasets, including adenoma, CRC and T2D. The FPR of these KO genes on non-IBD cohorts were also relatively low, with adenoma at 0.04, CRC at 0.13 and T2D at 0.07 (Supplementary Fig. 3e), indicating their excellent disease specificity.

We also utilized the EggNOG orthologous gene classification method for diagnosing IBD (Supplementary Fig. 3f and Supplementary Table 4), but the accuracy of IBD detection was slightly lower compared to using the KO models. The AUROC values ranged from 0.63 to 0.80 for cohort-to-cohort transfer validation (Supplementary Fig. 3g) and from 0.65 to 0.92 in LOCO validation (Supplementary Fig. 3h). After considering the accuracy and interpretability of the results, we decided to use only the annotated results from the KO database in subsequent analyses.

## Demonstration of Metabolomic Alterations across Different IBD Cohorts and Utilization of Signature Metabolites for IBD Diagnosis

Intrigued by the intricate interplay between gut microbiota and host co-metabolism[38], we further sought to explore the spectrum of changes in fecal metabolites through metabolomics. Using targeted metabolomics, we examined the differences in fecal metabolites between IBD patients and healthy individuals (Fig. 4a). We performed Principal Coordinate Analysis (PCoA) and constructed Orthogonal Partial Least Squares Discriminant Analysis (OPLS-DA) models, revealing substantial differentiation in the metabolomic profiles between the two groups (Fig. 4b, c and Supplementary Fig, 4a, b). Our findings indicate that no particular group of metabolites was superior in distinguishing between patients with IBD and healthy individuals than the collective set of all metabolites. (Supplementary Fig. 4c, d).

To further understand the differences in metabolites between IBD patients and healthy controls, we performed a differential analysis and identified 78 metabolites (Fig. 4d and Supplementary Tables 5). Most of these metabolites were found to be enriched in IBD patients, with only a few being depleted. The majority of these differential metabolites belong to the three major nutrient metabolism categories, including amino acids, carbohydrates, and fatty acids. Notably, amino acids such as Tryptophan, Glutamine, Arginine, and 5-Hydroxytryptophan, Histidine were found to be enriched in the intestines of IBD patients, this is consistent with the results of the previous studies[39]. Interestingly, we also observed that various organic acids related to the tricarboxylic acid cycle, such as Pyruvic acid, Fumaric acid, Malonic acid, and Oxoglutaric acid, were enriched in IBD patients, indicating abnormal energy metabolism of intestinal microbiota. Furthermore, among these 78 differential fecal metabolites, 36 are unique to the in-house dataset (Supplementary Fig. 4e), such as some amino acids (1-Methylhistidine, Acetylglycine, N-Acetylglutamine, N-Acetylserine Dimethylglycine and 4-Hydroxyproline) and benzenoids (Phenylpyruvic acid, 3-Hydroxyphenylacetic acid, Protocatechuic acid and 3-Aminosalicylic acid). Especially, carnitines compounds and 1-Methylhistidine are biomarkers associated with meat consumption[40] and its significant elevation in the gut of IBD patients is in line with well-established dietary risk factors for IBD, such as the consumption of red and processed meats[41]. However, the roles of these metabolites in IBD remain unknown.

We next aimed to investigate whether a specific group of metabolites can serve as an accurate diagnostic tool for IBD, regardless of whether it is targeted or non-targeted datasets through CCIA methods. We then integrated four metabolomics studies and identified a total of 79 metabolites that were commonly present in all four cohorts using HMDB ID. Subsequently, we conducted univariate differential analysis with FDR < 0.0001 and OPLS-DA analysis using a VIP score > 1, which led to the identification of 32 candidate metabolites. To further refine our selection, we employed IFE and narrowed down our pool to 13 metabolites (Fig. 4e, f). To account for variations in metabolite detection methods and numerical units between internal and external cohorts, we limited our cross-validation to cohorts with the same units. In our analysis of the Renji cohorts, the 10-fold cross-validation of our model achieved an AUROC value of 0.945, while the leave-one-out cross-validation (LOOCV) had a slightly lower performance of 0.937, and the independent validation cohort arm an AUROC of 0.867 (Fig. 4g). In the USA-NL cohorts, both 10-fold cross-validation and LOOCV exhibited similar AUROC values, both exceeding 0.9, and the independent validation cohort had good performance with an AUROC of 0.841 (Fig. 4h). Based on these results, it can be inferred that metabolomics has a higher potential for disease diagnosis compared to metagenomics and can be a promising biomarker for diagnosis in the future.

To further validate the disease specificity of our feature metabolites, we incorporated four additional non-IBD metabolomics cohorts, including one adenoma cohort, two CRC cohorts, and one T1D cohort. Differential analysis revealed that the majority of the 13 metabolites we included for diagnosing IBD did not show significant differences in these cohorts (FDR > 0.05, Supplementary Fig. 4f-i). These findings substantiate the disease-specific nature of our featured metabolites in IBD.

## Construction of multi-omics biological correlation (MOBC) maps of gut microbiota in IBD

Gut metabolites and microbiota are closely associated, however, fecal metabolite data presents a complex mixture derived from the host, gut microbes, and ingested food, emphasizing the urgent need to identify the specific metabolites driven by the gut microbiota. Therefore, we constructed an analytical framework - the multi-omics biological correlation (MOBC) maps, utilizing fecal metagenomic and metabolomic data from four cohorts ($n = 391$, Fig. 5a). We utilized the KEGG database to establish links between the 32 previously identified differential metabolites and the genes responsible for encoding potential enzymes directly involved in their metabolic processes (Fig. 4e). This effort led to the identification of 736 KO genes (Supplementary Table 6). Additionally, we conducted an intersection analysis, comparing these metabolic-related genes with the 162 KO genes identified in our previous differential analysis of 6 IBD metagenomic cohorts (Fig. 3a). As a result, our investigation unveiled 8 differential KO genes with associations to both metabolism and disease phenotypes. (Fig. 5b and Supplementary Fig. 5a-h). Further research on these genes and their roles in metabolic pathways could provide significant insights into the underlying mechanisms of IBD.

The K22477 (*argO*, N-acetylglutamate synthase) is responsible for producing N-acetylglutamate (NAG) from glutamate and acetyl-CoA.

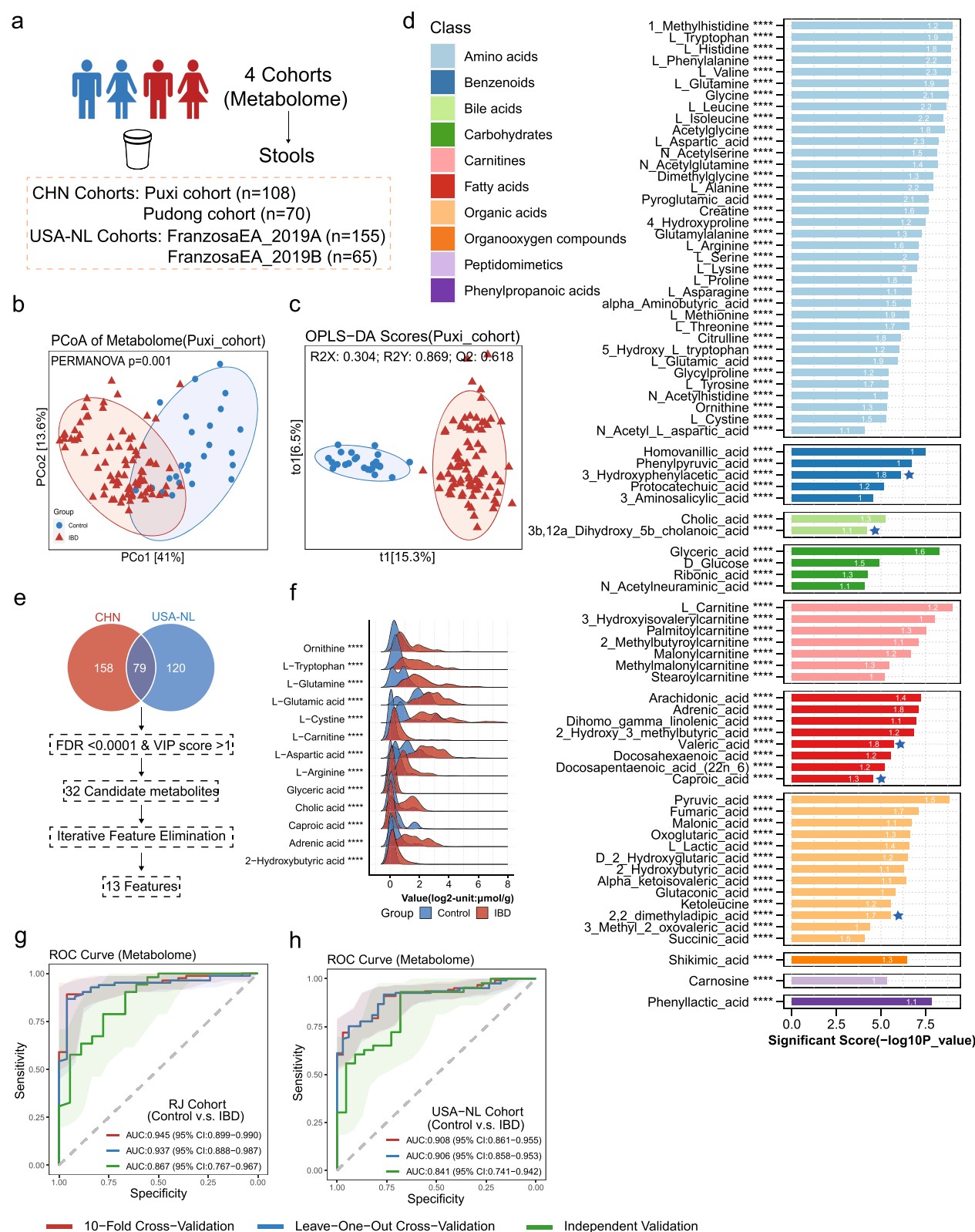

Our study revealed that IBD patients have reduced levels of K22477, leading to an excess of L-glutamate. This overabundance of L-glutamate can cause symptoms such as abdominal pain and diarrhea in IBD patients[42](Fig. 5c). Additionally, the lack of K22373 (*larA*, lactate racemase) prevents the conversion of (S)-lactate to (R)-lactate, which may contribute to the development of intestinal inflammation and malignant transformation[43](Fig. 5d). The K00290 (*LYS1*, saccharopine

dehydrogenase) reduces the likelihood of oxidative stress, a hallmark of inflammation in the gut affected by IBD[44](Fig. 5e). However, IBD patients with reduced levels of *LYS1* are unable to support a healthy microbiota due to the resulting oxidative stress. Urea plays a complex role in intestinal diseases, and in IBD patients, dysbiosis can result in abnormal urea metabolism, causing damage to the intestinal mucosal barrier and exacerbating inflammation[45]. Our study suggests that a

**Fig. 4 | Demonstration of Metabolomic Alterations across Different IBD Cohorts and Utilization of Signature Metabolites for IBD Diagnosis. a** Overall composition of the population across 4 metabolomics datasets ($n = 398$). **b** A principal coordinates analysis (PCoA) was performed on individuals from the Puxi cohort. The analysis showed significant differences in metabolites composition between control ($n = 25$) and IBD ($n = 83$) ($P = 0.001$), as determined by PERMANOVA using Bray-Curtis distance with 999 permutations (two-sided test). **c** Orthogonal Partial Least Squares Discriminant Analysis (OPLS-DA) model of Puxi cohort individuals based on gut metabolomic profiles (Q2Y: 0.618). The full model's predictive performance is evaluated using the cumulative Q2Y metric: Q2Y ranges from 0 to 1, and the higher the Q2Y, the better the performance. OPLS-DA model validation utilizes a permutation test (perml = 200, two-sided test), which is a nonparametric test. **d** The presented bar graph depicts metabolites that demonstrate significant differences between normal controls and IBD, with their significance scores calculated using a two-sided Wilcoxon test with FDR-corrected P values (cut-off value: $P < 0.0001$, denoted as ****). Each bar on the graph shows a white number that signifies the VIP score of the OPLS-DA model. The colors of the bars represent various categories of metabolites, with blue star-marked bars indicating metabolites that are depleted in IBD, while unmarked bars represent enriched metabolites in IBD. Exact p values are provided in the Source data file (Supplementary Table 5). **e** The workflow for identifying metabolites across four cohorts involved annotating with the Human Metabolome Database (HMDB). **f** The ridge plot shows the concentration differences of 13 featured metabolites between the normal control and IBD (Unit: $\log_2 (\mu mol/g)$). Their significance scores calculated using a two-sided Wilcoxon test with FDR-corrected $p$ values (denoted as ****$p < 0.0001$). Exact p values are provided in the Source data file. **g, h** We developed random forest (RF) classifiers trained on metabolites to identify IBD patients. We performed 10-fold cross-validation (red), leave-one-out cross-validation (LOOCV) (blue) and independent validation (green). Shaded areas represent the 95% confidence intervals of the corresponding ROC curves. Source data are provided as a Source Data file.

reduction in the symbiotic bacterial community involved in urea metabolism leads to the downregulation of K01476 (*rocF*, arginase), ultimately resulting in urea accumulation (Fig. 5f). These findings indicate that the gut microbial biotransformation is impaired in patients with IBD, resulting from a substantial reduction in the gut commensal community (Fig. 2e).

Furthermore, we identified four genes that encoding aminoacyl-tRNA biosynthesis enzymes within the symbiotic bacterial community, namely K09759 (*aspS*, nondiscriminating aspartyl-tRNA synthetase), K01870 (*ileS*, isoleucyl-tRNA synthetase), K01892 (*hisS*, histidyl-tRNA synthetase), and K01873 (*valS*, valyl-tRNA synthetase) (Fig. 5g, j). This is consistent with previous results in KEGG Orthology Enrichment Analysis which showed a significant increase in aminoacyl−tRNA biosynthesis pathway (Supplementary Fig. 3a). Aminoacyl-tRNA synthetases (ARSs) play an indispensable role in protein synthesis. Recent studies indicate that these enzymes encompass biological functions that extend beyond translation[46–48]. It has been reported that the gut-associated bacterium *Akkermansia muciniphila* (Am) secretes seryl-tRNA synthetase (AmTARS), which could modulate immune homeostasis and facilitate the production of anti-inflammatory IL-10[49]. These studies indicate that aminoacyl-tRNA synthetases from the gut microbiota may play a crucial role, potentially influencing the host immune and regulating gut homeostasis.

### Multi-omics signatures integration for diagnosing IBD across different cohorts

We have previously discovered three panels, consisting of 31 species, 25 KO genes, and 13 metabolites, that were able to accurately distinguish between IBD patients and normal controls. To investigate if the integration of multiple data sources could improve diagnostic accuracy, we further examined the interplay between gut microbiota and their metabolites. We first combined species and KO genes to differentiate IBD and obtained a satisfactory diagnostic performance in both 10-fold cross-validation and LOOCV. The Renji cohort showed an AUROC value above 0.97, while the USA-NL cohort increased to above 0.9 (in comparison to using single species or KO genes), and the independent validation of these panels exhibited AUROC values above 0.9 (Fig. 6a, d). After that, we combined species and metabolites, metabolites and KO genes, and found that their diagnostic performance was generally above 0.9 (AUROC) (Fig. 6b, c, e, f). Particularly, the combination of species and metabolites presented the best performance among the combined panels, and both outperformed the diagnostic performance of the individual panels.

After achieving good diagnostic performance in the above models, we further explored whether combining the all panels could enhance the diagnostic performance of our model. To our surprise, we found that the combined features significantly improved the diagnostic performance of our random forest model. In the Renji cohort, 10-fold cross-validation achieved an AUROC value of 0.98, while independent validation reached 0.96 (Fig. 6g). In the USA-NL cohorts, 10-fold cross-validation could reach 0.93, and independent validation could reach 0.92 (Fig. 6h). Our results indicate that combining species, KO genes, and metabolites can significantly enhance the diagnostic performance of our model in fecal metagenomic and metabolomic analysis.

### Identification of multi-omics biomarkers for distinguishing subtypes of IBD

As the treatment strategies for UC and CD differ significantly, we next aimed to identify a subset of markers from the multi-omics panel mentioned above that could distinguish between the two subtypes of IBD. Using the IFE approach, we selected 12 features from the multi-omics panel (Supplementary Fig. 6a). The RF model revealed that the 12 selected markers could effectively differentiate between UC and CD. The AUROC values for 10-fold cross-validation and LOOCV were approximately 0.8 in both internal and external cohorts, with independent validation achieving values above 0.7 (Supplementary Fig. 6b, c). This sub-panel can aid clinicians in further categorizing the disease following the diagnosis of IBD.

### Discussion

Although previous studies has utilized fecal biomarkers for diagnosing IBD[9,10], there are still two unresolved issues: the reliable reproducibility of biomarkers obtained from fecal samples across different cohorts and populations, and whether it's possible to further enhance the diagnostic performance of the existing fecal diagnostic model.

In this study, we integrated metagenomic and metabolomic data from multiple cohorts to identify 31 species, 25 KO genes and 13 metabolites distinguishing normal control from IBD cases. These biomarkers demonstrate robust reproducibility across various cohorts. In contrast to the current invasive gold standard for diagnosing IBD, colonoscopic examination, we have demonstrated the potential of utilizing gut fecal microbiota and metabolites as a non-invasive approach to diagnosis, which contributes to the early detection and prediction of IBD, facilitating timely interventions and reducing the risk of complications. Moreover, through the integration of diverse omics data, we have achieved a significant enhancement in the performance of our machine learning models for diagnosing IBD, achieving an AUROC value of 0.98 in Renji cohorts, compared to previous non-invasive diagnostic models[9,10]. Furthermore, we have successfully pinpointed a subset of markers from the multi-omics panel that exhibit the remarkable ability to distinguish between UC and CD with an impressive average AUROC value of up to 0.8. This discovery offers a promising potential for a non-invasive biomarker that could play a pivotal role in clinically categorizing distinct subtypes of IBD.

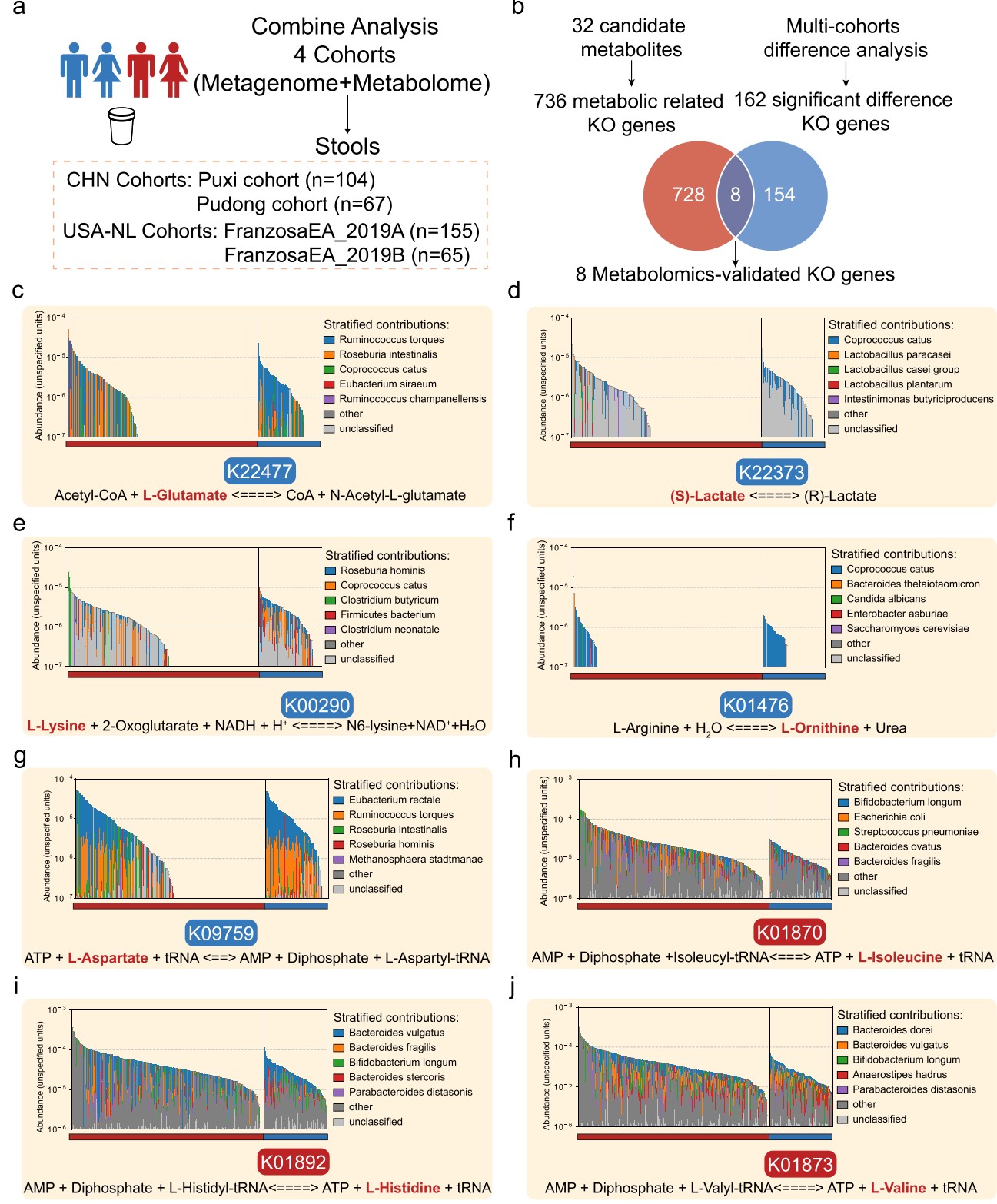

**Fig. 5 | Construction of multi-omics biological correlation (MOBC) maps of gut microbiota in IBD. a** 4 IBD cohorts are available for integrative analysis ($n = 391$). **b** Flowchart for the identification of 8 different KO genes related to metabolism and disease phenotypes to construct multi-omics biological correlation (MOBC) maps. **c–j** MOBC maps of the 8 KO genes. The reaction equation below each image represents a reaction process corresponding to a KO gene. The bolded metabolites and KO genes represent those that have been validated and show significant differences in our metagenomics and metabolomics analyses. The red color indicates enrichment in IBD, while blue represents depletion. The bar chart in the top half of each image represents the top 5 contributors of gut bacteria carrying this KO gene. Source data are provided as a Source Data file.

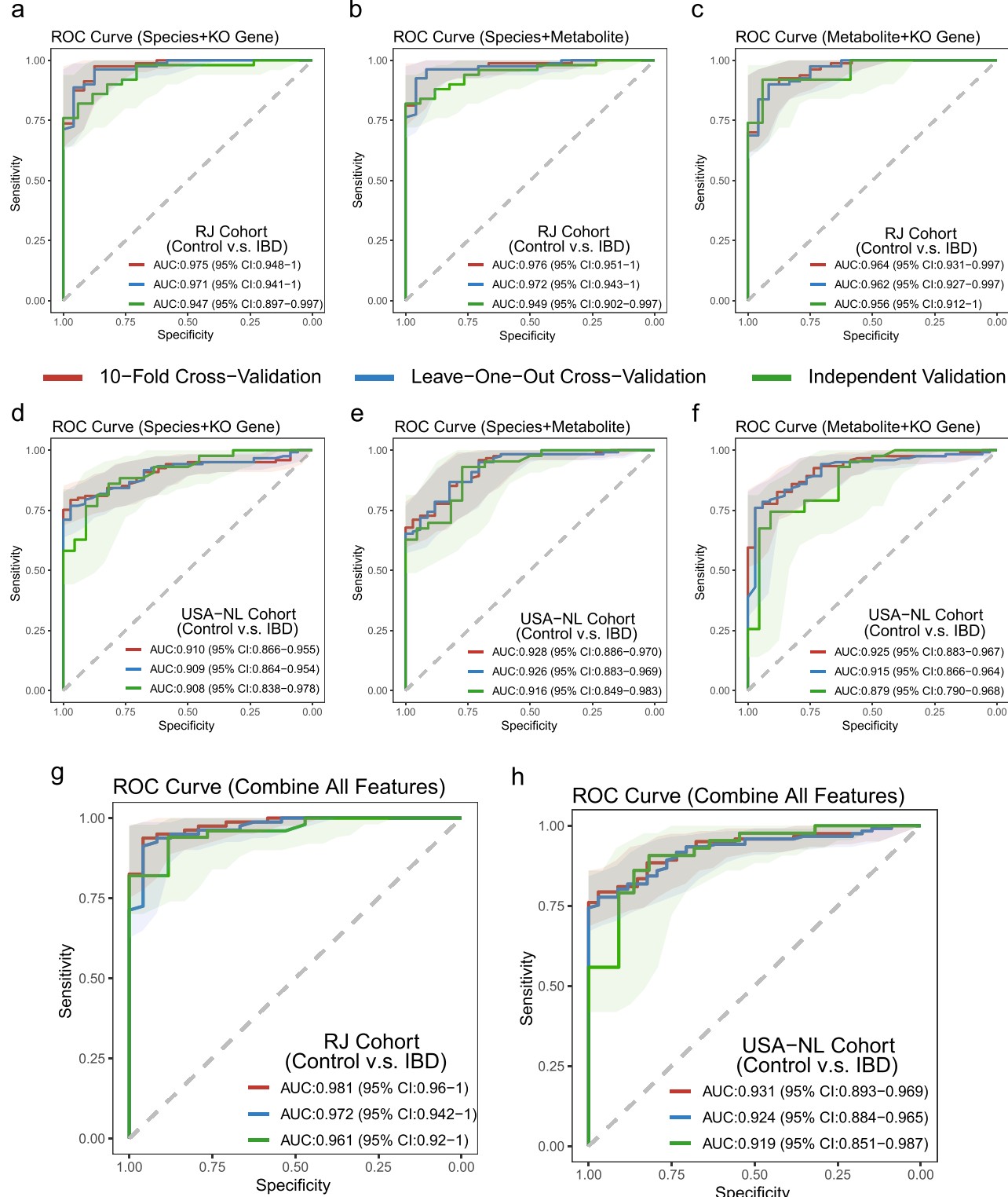

**Fig. 6 | Multi-omics signatures integration for diagnosing IBD across different cohorts.** We developed Random Forest (RF) classifiers to identify patients with IBD using multi-omics data. Specifically, we trained three different RF classifiers: one on species and KO genes (**a, d**), one on species and metabolites (**b, e**), and one on metabolite and KO genes (**c, f**). The training and testing of these classifiers were carried out using 10-fold cross-validation (red) and leave-one-out cross-validation (LOOCV) (blue) within the Puxi or FranzosaEA 2019A cohorts, respectively. The performance of these classifiers was then validated on independent validation sets (green) in the Pudong or FranzosaEA 2019B cohorts. In addition, we also trained RF classifiers using a combined panel of metabolites, species, and KO genes (**g, h**). Shaded areas represent the 95% confidence intervals of the corresponding ROC curves.

In addition, previous studies have shown that differences in gut bacteria or metabolites found in different case-control studies mostly indicate a general imbalance in the gut ecosystem[50], rather than specific changes linked to certain diseases. This highlights the difficulty in identifying distinct patterns of gut bacteria or metabolites that are specific to particular diseases. In our present study, we successfully formulated disease-specific signatures that exhibited a low false positive rate across gastrointestinal (GI) conditions, such as adenoma and colorectal cancer (CRC), as well as non-GI diseases like diabetes. In summary, this multi-omics model can propel innovations in both clinical practice and the realm of medical science.

While unresolved causality among microbial, metabolite, and host processes during IBD development is not a primary focus for diagnostic purposes, elucidating the underlying mechanisms would significantly enhance our understanding of this disease. To achieve this objective, we have developed comprehensive workflows for both functional metagenome and metabolome analysis. Firstly, through the analysis of functional gene (KO genes) within the gut microbiota, we identified multiple regulatory pathways associated with disease development involving the gut microbiota. Notably, the Two-component systems, composed of sensor histidine kinases and response regulator proteins, play a crucial role in bacterial and archaeal signal transduction processes[34]. Additionally, we discovered a key transcriptional regulatory factor enriched in this pathway, known as *crp* (Cyclic AMP Receptor Protein). *crp* belongs to the CRP-FNR superfamily of transcription factors and is activated as a DNA-binding protein by binding with its allosteric effector, cAMP[51]. The regulation of the Two-component system or *crp* holds promise as a therapeutic target for treating IBD. Through MOBC maps, our research has uncovered a significant enrichment in the Aminoacyl-tRNA biosynthesis pathway. Aminoacyl-tRNA synthetases (ARSs) play a vital role as catalysts in protein synthesis across all living organisms. However, their functions have evolved over time, and a growing body of research indicates that their non-classical functions might hold even greater importance[46–49,52]. The diverse functionality of ARSs has revealed their potential as a valuable and underutilized resource for therapeutic targets in IBD.

Of course, our study does have some limitations. Being a cross-cohort study, it's difficult to completely eliminate biases in areas like selecting cohorts, collecting samples, and analyzing methods. However, we've taken extensive measures to minimize these factors, ensuring the study's scientific rigor. Still, there could be unknown factors influencing the results, such as diet, medication, and lifestyle choices, which require further investigation for validation.

In conclusion, our study revealed the overall patterns of changes in the gut microbiome and metabolome of IBD patients using CCIA. This information can be valuable for exploring interventions and treatments for IBD. Further research should validate and delve deeper into the underlying molecular mechanisms, such as longitudinal studies and interventions. While microbiota and metabolite interventions hold promise for managing health conditions, they also come with challenges and ethical considerations that need careful evaluation, for example, microbiota variation, long-term effects, and standardization need thorough attention. Longitudinal studies are crucial to fully understand how the gut microbiota and metabolome change during different stages of IBD and in response to interventions.

## Methods

### Ethics statement

The patient cohorts were approved by the ethics committee of Renji Hospital affiliated to the School of Medicine, Shanghai Jiao Tong University, China, the ethical approval number are 2019-qkwkt-001 and 2021-skt-004. In this study, we did not have any specific requirements regarding the participants' gender, and the gender of participants was determined based on self-report. All participants provided informed consent prior to their inclusion in the study.

### Participants enrollment

In this study, we recruited two IBD cohorts from Renji Hospital, Shanghai, including the Puxi and Pudong campuses, for the discovery and validation cohorts, between between January 1, 2019, and December 31, 2022, respectively. We also recruited a group of healthy control subjects who were carefully matched by age and gender across two hospital campuses. It should be noted that all participants who were enrolled provided informed consent. The enrollment was followed the specific inclusion and exclusion criteria, which are provided in the follow.

The inclusion criteria included: (1) Participants must be aged between 16 and 65 to be eligible. (2) IBD group were patients newly diagnosed with UC or CD by combining clinical symptoms, imaging, endoscopic and pathological appearances, and had not received any treatment the time of enrollment; (3) Control group was a healthy control population that did not have any significant abnormality in colonoscopy; (4) the participants were capable of understanding and completing the questionnaire, and were willing to cooperate in the collection of fecal samples and basic and clinical information. The exclusion criteria include: (1) medication history of antibiotics, probiotics, immunosuppressants, hormone, or non-steroidal anti-inflammatory drugs within three months before enrollment; (2) abdominal surgery history within six months before enrollment; (3) history of cancer, other autoimmune disease excluding IBD, organ transplantation, or other serious digestive diseases; (4) uncontrolled systemic metabolic disorders such as blood pressure, blood glucose, blood lipids within six months before enrollment; (5) severe and uncontrolled gastrointestinal symptoms such as severe gastrointestinal bleeding, severe diarrhea, severe constipation, gastrointestinal obstruction, etc., within six months before enrollment; (6) significant changes in dietary habits, such as the initiation of a vegan diet, etc., within six months before enrollment; (7) inability to cooperate or unwillingness to cooperate with this study.

In the metagenomic cohorts (Fig. 2a), a total of 208 participants were enrolled, comprising 138 patients diagnosed with IBD and 70 healthy control subjects, matched for age and gender. Specifically, the Puxi cohort ($N = 132$, control=45, IBD = 87) and the Pudong cohort ($N = 76$, control=25, IBD = 51) were employed for both model discovery and validation purposes. For the metabolomic cohorts (Fig. 4a), a total of 178 participants were included, with 135 individuals diagnosed with IBD and 43 healthy control subjects, carefully matched for age and gender. Among these, the Puxi cohort ($N = 108$, control=25, IBD = 83) and the Pudong cohort ($N = 70$, control=18, IBD = 52) were utilized for both model discovery and validation phases. In the combined analysis cohorts (Fig. 5a), a total of 171 participants were incorporated, consisting of 130 patients with IBD and 41 age- and gender-matched healthy control subjects. Among them, the Puxi cohort ($N = 104$, control=24, IBD = 80) and the Pudong cohort ($N = 67$, control=17, IBD = 50) were utilized for both model discovery and validation stages. The details of recruitment for the in-house IBD Renji cohorts are shown in Supplementary Fig. 1a. The clinical characteristics of the study participants are shown in Supplementary Table 1.

### Public cohorts of patients with IBD and normal controls

The metagenomic raw sequencing data for the FranzosaEA 2019A and FranzosaEA 2019B cohorts (PRJNA400072), and HeQ 2017 (PRJEB15371) were downloaded from the European Nucleotide Archive. In addition, the metagenomic sequencing data for the HallAB 2017 cohort (PRJNA385949), NielsenHB 2014 cohort (PRJEB1220), HMP 2019 ibdmdb cohort (PRJNA398089) and the LifeLD VilaAV 2018 cohort (EGAS00001001704, EGAD00001004194) data were acquired from the curatedMetagenomicData. The two published metabolomics cohorts were obtained from a previously published study[10] and contained pre-processed data. All metadata for both metagenomics and

metabolomics were manually curated from the materials provided in published papers.

## Stool sample collection

All participants are required to provide a minimum of 3.0 grams of stool sample upon enrollment. Samples are to be collected in a sterile specimen collector (Thermo Scientific, USA, R21922) provided by the investigator in advance. After collection, the samples must be promptly transferred to a −80 °C ultra-low temperature cryogenic freezer for storage within 4 hours, pending further processing in three months.

## Study design and Sample filtering

We included a total of nine metagenomic datasets, consisting of 6 discovery cohorts and 3 independent verification cohorts (Fig. 2a), as well as four metabolomics datasets, including two targeted metabolomics datasets and two untargeted metabolomics datasets (Fig. 4a). Among these datasets, four datasets contained both metagenomic and metabolomic data (Fig. 5a). Considering that some subjects were sampled at different time points, we only retained the data from the first sampling to ensure the accuracy of the diagnostic model. Additionally, as country or region is a major confounding factor, we only included subjects from the same country in each dataset to minimize confounding effects. Furthermore, because there were significantly more normal controls than IBD cases in the LifeLD VilaAV 2018 cohort, we randomly removed some of the normal controls to maintain a ratio of 1:2 to 1:3 between normal controls and IBD cases for precision of the study.

## Metagenomic sequencing

The DNA from the stool samples was extracted utilizing the HiPure Stool DNA Mini Kit (Magen Biotechnology, China). The quality, size, and concentration of the extracted DNA were evaluated via agarose gel electrophoresis and the Qubit™ 4 Fluorometer (Thermo Fisher Scientific, USA). The metagenomic libraries were subsequently constructed by NeoBIO techology utilizing the Hieff NGS® Ultima DNA Library Prep Kit for Illumina® (Yeasen Biotechnology, China) following the manufacturer's protocol. After ensuring the quality of the libraries, high-throughput sequencing was performed on the NovaSeq6000 platform (Illumina, USA).

## Metabolite quantification

Targeted metabolomics profiling was conducted using the Q300 Metabolite Array Kit from Metabo-Profile Biotechnology of China (Xie et al. [53]). In brief, to extract metabolites from lyophilized feces, a homogenate was prepared using 10 mg of feces with 25 µL of water. The mixture was then extracted with 185 µL of cold ACN-Methanol (8/2, v/v) and centrifuged. Next, 30 µL of the supernatant was derivatized with 20 µL of freshly prepared derivative reagents on a Biomek 4000 workstation. Internal standards were added to the derivatized samples, which were then randomly analyzed and quantitated using an ultra-performance liquid chromatography coupled to tandem mass spectrometry (UPLC-MS/MS) system. A total of 310 standard substances, including 12 subclasses, were obtained from Sigma-Aldrich, Steraloids Inc, and TRC Chemicals. To ensure the quality of the metabolomics platform, three types of quality control samples were routinely used: test mixtures, internal standards, and pooled biological samples. The derivatized pooled quality control samples were injected every 14 test samples (Supplementary Fig. 4a). The raw data generated by UPLC-MS/MS were processed using the QuanMET software (v2.0, Metabo-Profile, Shanghai, China) for peak integration, calibration, and quantification of each metabolite. Through mass spectrometry-based quantitative metabolomics, metabolomic features were annotated to metabolites with Level 1 of confidence by comparing them to the standard metabolites.

## Metagenomic profiling

The study utilized the bioBakery meta-omics workflow to generate taxonomic and functional profiles from metagenomic data. To ensure the use of high-quality microbial reads free from contaminants, KneadData was employed for data filtering. Taxonomic profiling was performed using MetaPhlan3, which utilizes a library of clade-specific markers to provide pan-microbial profiling. Functional profiling was conducted using HUMAnN3, which constructs a sample-specific reference database from the pangenomes of the subset of species detected in the samples by MetaPhlAn3. To quantify gene presence and abundance on a per-species basis, sample reads are mapped against this database. In cases where reads fail to map at the nucleotide level, a translated search is conducted against a UniRef-based protein sequence catalogue to identify gene families (UniRef90s). The resulting abundance profiles are stratified by each species contributing to those genes and further summarized into higher-level gene groups such as KOs (KEGG Orthologs) and EggNOGs (Evolutionary Genealogy of Genes: Non-supervised Orthologous Groups).

## Preprocessing of taxonomic abundance profiles and functional abundance profiles

To ensure the accuracy and reliability of statistical analysis on metagenomic data, it is essential to undertake stringent data filtering measures to reduce data noise and enhance data quality. One crucial step is the removal of low-abundance microorganisms or genes, which exhibit low expression levels in the sample and may represent contaminants or batch effects in the environment. Similarly, entities with no variance must be filtered out before analysis. This approach can minimize the impact of technical noise or experimental errors, resulting in improved reproducibility and greater stability of experimental results.

Subsequently, a pseudo-count of $1\times10^{-5}$ was added to avoid non-finite values resulting from log10(0), and the abundances were log10-transformed. To prepare functional profiles, such as EggNOG genes or KO genes abundance profiles, the same preprocessing steps were applied as for the species profiles, as previously described. However, for these functional profiles, a maximum abundance cutoff of $1\times10^{-6}$ was used, and a pseudo-count of $1\times10^{-9}$ was added during the log transformation. Finally, to obtain standardized values, the abundances profiles were converted into z-scores.

## Confounder analysis

We conducted an ANOVA-type analysis to determine the impact of potential confounding factors on individual microbial species, relative to the impact of IBD. A linear model was employed, which included both IBD status and the confounding factor as explanatory variables for species abundance. The analysis assessed the total variance within the abundance of a specific microbial species relative to the variance explained by disease status and the variance explained by the confounding factor. Overall, this methodology facilitated a more comprehensive assessment of the factors that influence the microbiome and their potential impact on disease outcomes.

## Microbial ecological analysis

Alpha diversity metrics, namely the Shannon and Simpson Indices, were utilized to evaluate the diversity and evenness of species within a community. The primary objective was to investigate the differences in alpha diversity between IBD and control cases using the statistical tool MMUPHin. In this analysis, the cohort was treated as the independent variable, while potential confounding factors such as gender and age were accounted for as random effects. This approach facilitated a more precise evaluation of the relationship between IBD and alpha diversity. Beta diversity was evaluated by computing the Bray-Curtis distance, which quantifies the dissimilarity of microbial communities across samples. To investigate the differences in microbial community

composition between disease groups or cohorts, a permutational multivariate analysis of variance (PERMANOVA) was employed using 999 permutations.

### Differential abundance analysis to identify gut microbial species and functional genes

The significance of differential abundance (DA) between different groups was evaluated using the 'coin' package in R and a blocked Wilcoxon test. Each species or gene was tested separately, and the data were blocked by cohort to control for any confounding effects that may have arisen from differences in cohort composition. To account for variations in block size and composition, permutations were performed within each block to obtain a conditional null distribution. To account for multiple hypothesis testing, p-values were adjusted using the FDR method. Additionally, we used a generalized fold change (gFC) approach to calculate the magnitude of differences between control and IBD samples. KEGG Orthology Enrichment Analysis (KOEA) were conducted for KO genes using the R package "clusterProfiler"[54].

### Identification of IBD-related differential metabolites

We identified differential metabolites based on two criteria: (1). False discovery rate (FDR) < 0.0001, using nonparametric univariate method (Wilcoxon rank-sum test). The $P$ value for each metabolite was corrected for FDR using the false discovery rate method (2). Significance of the projected variable (VIP score) > 1 using OPLS-DA model. OPLS-DA stands for Orthogonal Projections to Latent Structures Discriminant Analysis, which is a multivariate statistical method used to analyze data with multiple variables. The significance of the projected variable is a measure of how well a metabolite contributes to the separation between the two groups being compared. The model validation of OPLS-DA utilizes a permutation test, relying on the "ropls" R package (Supplementary Fig 4b, d).

### Preprocessing of metabolomics profiles

Given the significant variability in metabolomics technology, processing methods, and output results among studies, we preprocessed our metabolomics dataset to facilitate subsequent cross-cohort analysis. To accomplish this, we utilized the MetaboAnalyst (5.0) compound ID conversion program to standardize metabolite names from both internal and external cohorts to a common HMDB ID and identified 79 metabolites that were common across all four cohorts. Subsequently, we applied a $\log_2$ transformation to the metabolite values and then converted them into z-scores.

### Iterative feature elimination

To improve the reliability and robustness of our model and reduce its size and complexity, we used the Iterative Feature Elimination (IFE) feature selection method in Python[55]. First, we performed a differential feature analysis to identify potential features. Subsequently, we utilized the scikit-learn package to train a random forest (RF) model, which was then subjected to stratified 10-fold cross-validation to distinguish between IBD and normal controls. We implemented stratified 10-fold cross-validation to allocate training and testing datasets appropriately. Next, we applied the Iterative Feature Elimination (IFE) step to enhance the performance of subsequent RF models. Finally, we selected the top features from the best-performing model (the model with the highest AURCO value) as the final features for modeling.

### Multiomics statistical modeling workflow and model evaluation

Given the strong performance of the random forest model, an ensemble machine learning approach, in the microbial data classification, our machine learning model also employs this model. First, we used 10-fold cross-validation within the cohort, a commonly used technique in machine learning and statistical analysis. This involved splitting the available data into 10 equal parts, training the model on

nine of these parts, and evaluating its performance on the remaining part. In cohort-to-cohort transfer validation, we trained the classifiers on a single cohort and tested them on all other cohorts. In leave-one-cohort-out (LOCO) validation, we set aside the data from one cohort as an external validation set and trained the model on the remaining data from all other cohorts. We then used the same nested cross-validation procedure as for cohort-to-cohort transfer validation. These methods allowed us to assess the generalizability of our metagenomic classifiers and their ability to perform well on data from multiple cohorts. The Leave-One-Out Cross-Validation (LOOCV) involves removing one sample (or observation) from the dataset, training the model on the remaining data. We then used the removed sample as the validation dataset to evaluate the model's performance. We repeated this process for each sample in the dataset, ensuring that each sample was used as a validation dataset exactly once. The data preprocessing, model building, and model evaluation were performed utilizing the following R packages: SIAMCAT (v.1.14.0), caret (v.6.0.90), randomForest (v. 4.7.1.1), pROC (v.1.18.0), and ROCR (v.1.0.11).

### Independent validation with external metagenomic cohorts

To ensure the reliability of metagenomic features as diagnostic markers for IBD, we validated our findings using three independent datasets from both the USA and China (Fig. 2a). We performed cohort-to-cohort and leave-one-cohort-out (LOCO) analyses to evaluate the strength and consistency of the identified markers, following the same process used to construct the model in the discovery cohorts.

### Validation of microbial biomarkers' specificity in non-IBD cohorts

To minimize the risk of misdiagnosing IBD, we assessed the specificity of metagenomic markers by analyzing the AUROC values of models constructed using the most effective panel of features. Our analysis included patients with non-IBD conditions such as colorectal cancer (60 cases and 65 controls from PRJEB27928, WirbelJ 2018 cohort), type 2 diabetes (45 cases and 39 controls from PRJEB1786, KarlssonFH 2013 cohort), and adenoma (47 cases and 61 controls from PRJEB7774, FengQ 2015 cohort).

### Validation of the specificity of metabolic biomarkers in non-IBD cohorts

In order to validate the disease specificity of our selected feature metabolites, we integrated four additional metabolomics cohorts, which comprised one adenoma cohort (KIM ADENOMAS 2020[56], $n = 204$), two CRC cohorts (KIM ADENOMAS 2020[56], $n = 138$ and YACHIDA CRC 2019[57], $n = 347$), and one T1D cohort (KOSTIC INFANTS DIABETES 2015[58], $n = 103$). All the data is sourced and available from the study by Muller, E. et al.[59].

### Statistics & Reproducibility

We did not employ a statistical method to predetermine the sample size because this analysis is based on a comprehensive examination of public data with a sufficient number of samples. To ensure the accuracy of the diagnostic model, we only retained data from the initial sampling, considering that some subjects were sampled at different time points. Moreover, as country or region is a major confounding factor, we only included subjects from the same country in each dataset to minimize confounding effects. Additionally, due to a significant imbalance between normal controls and IBD cases in the LifeLD VilaAV 2018 cohort, we randomly excluded some normal controls to maintain a ratio of 1:2 to 1:3 between normal controls and IBD cases, enhancing the precision of the study. The experiments were not randomized as statistical analyses relied on disease status information. Data collection and analysis were not conducted blind to the experimental conditions. Given the non-normally distributed nature of microbial data, relevant statistical

analyses were performed using non-parametric tests, such as the Wilcoxon signed-rank test.

## Data availability

The metagenomics data generated in this study have been deposited in the China National Center for Bioinformation database under accession code PRJCA017408. Additionally, all other sequencing data analyzed in this work are available in public databases, including the curatedMetagenomicData (PRJNA385949, PRJEB1220, PRJNA398089, EGAS00001001704, EGAD00001004194, PRJEB27928, PRJEB1786, PRJEB7774, https://bioconductor.org/packages/curatedMetagenomic Data) and the European Nucleotide Archive (PRJNA400072, PRJEB15371, https://www.ebi.ac.uk/). The metabolomics mass spectral raw data generated in this study have been deposited in MetaboLights under accession code MTBLS8713 (www.ebi.ac.uk/metabolights/ MTBLS8713). The metabolomics data from the external cohorts are sourced from the supplementary materials of their respective articles[10]. The metabolomics data of non-IBD cohorts are sourced from the study by Muller, E. et al. [59] (https://github.com/ borenstein-lab/ microbiome-metabolome-curated-data). The Human Metabolome Database (HMDB) is a freely accessible electronic database that provides comprehensive information about small molecule metabolites found in the human body (https://hmdb.ca/). Source data are provided as a Source Data file. Source data are provided with this paper.

## Code availability

The software packages used in this study are free and open source. The bioBakery tools (KneadData, MetaPhlAn3 and HUMAnN3) used to process multi-omics sequencing data are available via http:// huttenhower.sph.harvard.edu/biobakery as source code and installable packages. The code and analysis scripts of this study are available on Zenodo (https://doi.org/10.5281/zenodo.8432120).

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

## Acknowledgements

This work was supported by various funding sources. We are grateful for the funding provided by the National Natural Science Foundation of China (grant numbers 82230016, 82272979 to J.H., 82073115 to H.C., 82103246 to C.S., and 81973346 to Y.Z.), the Program for Professor of Special Appointment (Eastern Scholar no. 201268 to J.H.) at Shanghai Institutions of Higher Learning, the Shanghai Municipal Education Commission—Gaofeng Clinical Medicine Grant Support (grant numbers 20152512 to H.C., 20161309 to J.H.), the Shanghai Municipal Health Commission Talent Program (grant numbers 2022XD048 to J.H.), and the National Key Research and Development Plan of the Ministry of Science and Technology (grant numbers 2022YFE0125300 to H.C., 2020YFA0509200 to J.-Y.F). The authors would like to express their gratitude to the participating patients and all members of collaborating laboratories for their valuable contributions and insightful discussions related to this research project.

## Author contributions

The study was conceived and designed by H.C. and J.H. Patient enrollment, sampling, and clinical measures were carried out by Y.-L.Z., H.S., Y.Z., Z.W. and Z.C. The raw data was downloaded by H.C., L.N., B.X., Y.-L.Z., X.H., M.H., X.Z., J.D., Y.Z. and Y.M., and sequencing data from metagenomics and metabolomics were analyzed by L.N., Y.-L.Z., H.S., Y.Y. and T.T. The original draft was prepared by L.N., Y.-L.Z., H.S., Y.Z., and C.S., while J.-Y.F., H.C. and J.H. revised and edited the manuscript. All authors read, discussed, and approved the final version of the manuscript.

## Competing interests

The authors declare no competing interests.

## Additional information

¹State Key Laboratory of Systems Medicine for Cancer; Key Laboratory of Gastroenterology & Hepatology, Ministry of Health; Division of Gastroenterology and Hepatology; Shanghai Cancer Institute; Shanghai Institute of Digestive Disease; Renji Hospital, Shanghai Jiao Tong University School of Medicine. 145 Middle Shandong Road, Shanghai 200001, China. ²Department of Gastroenterology, Xuzhou Central Hospital, Clinical School of Xuzhou Medical University, Xuzhou, China. ³Department of Medical Oncology, Xuzhou Central Hospital, Clinical School of Xuzhou Medical University, Xuzhou, China. ⁴Department of Gastroenterology and Hepatology, Zhongshan Hospital, Fudan University, Shanghai, P.R. China. ⁵Department of Gastrointestinal Surgery, Renji Hospital, Shanghai Jiao Tong University School of Medicine. 145 Middle Shandong Road, Shanghai 200001, China. ⁶These authors contributed equally: Lijun Ning, Yi-Lu Zhou, Han Sun, Youwei Zhang, Chaoqin Shen. ✉e-mail: haoyanchen@sjtu.edu.cn; jiehong97@sjtu.edu.cn

