## [Peer Review File · Nature Communications]

REVIEWER COMMENTS

Reviewer #1 (Remarks to the Author):

This study assumes that there's large heterogeneity across studies and results are not consistent, limiting our ability to understand inflammatory bowel disease (IBD) and find diagnostic biomarkers. For this reason, the authors studied and integrated 9 metagenomic cohorts (divided into 6 discovery and 3 validation cohorts) and 4 metabolomic cohorts (divided into 2 external and 2 in-house cohorts) to study IBD.

Although the rationale behind it could be valid, the results here presented add no novelty to the literature and some claims are not supported by the data. The four main results are: (1) Depletion of microbiota in IBD; (2) Accumulation of amino acids in IBD; (3) functional dysbiosis of the microbiota in IBD; and (4) Abnormal (excessive) production of ATP in IBD.

When looking at figure 2E and the per-cohort p values the data looks quite consistent across individual cohorts. These are public cohorts, and some are published in peer-reviewed papers, so I miss a better interpretation of the integrative results with respect to those published as individual studies. What is the main gain of the integrative analysis? What are exactly the main microbiota taxa and metabolites that were not found in any of the individual studies?

For instance, gut microbiota depletion or reduced microbial diversity has been reported before in several publications. Amino acids have been associated before to IBD.

The authors try to assess the disease-specific microbiome signature by analyzing data from three other diseases: GI, CRC and T2D. However, the LOCO classification models seem biased and needs further validation, including metabolites. The metabolomic signature seems very unspecific and could be associated to other diseases.

The results from MOBC claiming abnormal ATP production by the microbiota in IBD patients is not supported by the data. Lines 374-384 are extremely speculative. There's no experimental evidence, neither from bacterial enzymes nor from the detected metabolites that could possibly indicate that the microbiota of IBM patients produces excessive amounts of ATP.

The study focuses on 79 metabolites shared by the four cohorts (2 external cohorts by non-targeted metabolomics, and 2 in-house cohorts by using targeted metabolomics); however it is unclear how

metabolites were annotated/identified in the non-targeted datasets. Were they annotated based on MS1 data alone, or MS1 and MS2 data? MS2 data and matching scores should be provided as supplementary information.

In general, the paper contains many vague interpretations in the results section.

Minor:

-Lines 290-293: revise the text, the style is excessively baroque for scientific writing

Reviewer #2 (Remarks to the Author):

General Comments:

The article presents a comprehensive analysis of multiple cohorts to investigate the role of gut microbiota and metabolome in inflammatory bowel disease (IBD). The study aims to eliminate biases and confounding factors, such as race and diet, to provide valuable insights for future interventions and treatments based on microbiota or metabolites for IBD. Overall, the article presents interesting findings, it is highly relevant for the field of IBD, adding novel insight compared to available literature. It is well designed, sound, with a strong methodology which is one of the major innovation of this paper. However there are a few areas that could benefit from further clarification and discussion.

Strengths :

- a. Large sample size: The inclusion of a substantial number of cases (1363 cases for metagenomic analysis and 398 cases for metabolomics analysis) enhances the reliability of the results.
- b. Comprehensive approach: The integration of metagenomic and metabolomics analyses, as well as the construction of Multi-Omics Biological Correlation (MOBC) maps, provides a comprehensive understanding of the relationship between gut microbiota, microbial functional genes, and intestinal metabolites in IBD.

Clarifications and Suggestions:

a. please specify better the clinical characteristics of patient enrolled into the selected cohorts, such as demographic information (e.g., age, gender) and disease-related characteristics (e.g., disease subtype, Montreal Classification, disease duration) of the IBD patients. These informations are crucially relevant to better interpret the clinical meaning of the results.

b. Biases and limitations: Acknowledge and discuss potential limitations of the study. Address the limitations related to the cohort selection process, sample collection, and analysis methods. Furthermore, discuss the potential impact of unmeasured confounders that might influence the observed associations.

Interpretation of Findings:

a. Mechanistic insights: Explain how future research could strengthen the findings of your study, postulating the kind of protocols and studies needed (i.e. longitudinal studies, interventional studies).

c. Diagnostic model: Provide more details regarding the development and validation of the diagnostic model. Discuss the potential clinical implications of the model and its performance in comparison to existing diagnostic methods.

Future directions:

a. Intervention and treatment strategies: Discuss how the identified alterations in gut microbiota and metabolome can guide the development of interventions and treatments for IBD. Are there any specific targets or pathways that appear particularly promising? Consider the potential challenges and ethical considerations associated with interventions based on microbiota or metabolites.

b. Longitudinal studies: Highlight the need for longitudinal studies to understand the dynamic changes in gut microbiota and metabolome during different stages of IBD and in response to interventions. This will provide insights into causality and the potential for personalized treatment approaches.

In summary, the article provides valuable insights into the role of gut microbiota and metabolome in IBD. Addressing the suggested clarifications and discussing the interpretation of findings in relation to existing knowledge will further strengthen the article's impact and contribute to the field of IBD research

Reviewer #3 (Remarks to the Author):

This is an interesting topic that covers the topic of microbiome and metabolomic changes in IBD. There are a number of issues with the manuscript as presented:

1. The number for ethical approval should be given in the manuscript.
2. The study describes enrolment of participants but is devoid of the basic reporting information associated with this. No information on numbers, timeframe enrolled, enrolment criteria are not adequately described. No description on how healthy controls were recruited.
3. Details describing the samples collection are inadequate: "Stool samples were aseptically collected into fecal collection tubes and immediately preserved at -80°C until further processing" What were the collection tubes. What time were the samples collected – how long before being processed for storage. Details of medication use are not clear – were they taken into consideration?
4. The details on the recruitment of participants etc do not have enough detail – the section on participant enrolment in the methods is devoid of detail. The registration of the study is not clear and the number of participants enrolled is not clear here or later in the paper. There are rigorous guidelines one should adhere to for the reporting of human participant data and this manuscript falls short. The reporting summary was not detailed enough.
5. Figure 1- provides a nice summary of the cohorts – but does it warrant figure 1 in the paper – more appropriate in the supplementary data.
6. The description of the metabolomic data is inadequate. For example metabolomic data is not sequenced yet the title is "metabolomic sequencing"
7. Not enough information is given in terms of the identification of the metabolites and the level of identification. Methods says QC samples were used but no information in relation to the QC data is presented.
8. The O-PLS-DA models for the metabolomic data needs to be validated – using permutation testing for example.
9. In general for the metabolomic data there is an over interpretation of the data – the data is faecal metabolite data which not only a representation of gut metabolites but instead has a complex mixture of metabolites originating from the host, gut microbes and ingested food. There is no appreciation of this in the manuscript and no attempt to differentiate metabolites that could potentially be microbe derived.
10. The novelty of the current work is not evident. There are previous results showing alterations in amino acids and keto acids in IBD patients. While cohorts is important work the novelty is lower due to previous published work in this area.

11. The results from the multiomic biological correlation over state that results – conclusions such as “The results from MOBC revealed functional impairments in gut microbial biotransformation and abnormal ATP production capacity in pathogenic bacteria” cannot be made without actual measurements of ATP or the functional capacity of the ATP production system. Changes in the TCA cycle intermediates cannot infer this alone.

Dear reviewers,

We are very grateful for the positive and comprehensive review, in which the referee and editors clearly noted the importance and the novelty of our work as well as their strong interest in our patient-oriented mechanistic and clinical studies in the paper. The reviewer provided a large number of valuable and useful suggestions. In responses, we only focus on his/her concerns, and present a revised version of the manuscript. The revised paper incorporates multiple new experiments now depicted in 6 figures with 39 subpanels, 6 supplementary figures with 43 subpanels, and 6 supplementary tables. Our intention is to properly match the referees' comprehensive review of our paper with equally comprehensive responses, and address each of the referee's collective concerns, in detail, below. Our point-to-point responses are as follows:

Reviewer #1:

Major concerns:

Comment 1: This study assumes that there's large heterogeneity across studies and results are not consistent, limiting our ability to understand inflammatory bowel disease (IBD) and find diagnostic biomarkers. For this reason, the authors studied and integrated 9 metagenomic cohorts (divided into 6 discovery and 3 validation cohorts) and 4 metabolomic cohorts (divided into 2 external and 2 in-house cohorts) to study IBD. Although the rationale behind it could be valid, the results here presented add no novelty to the literature and some claims are not supported by the data. The four main results are: (1) Depletion of microbiota in IBD; (2) Accumulation of amino acids in IBD; (3) functional dysbiosis of the microbiota in IBD; and (4) Abnormal (excessive) production of ATP in IBD. When looking at figure 2E and the per-cohort p values the data looks quite consistent across individual cohorts. These are public cohorts, and some are published in peer-reviewed papers, so I miss a better interpretation of the integrative results with respect to those published as individual studies. What is the main gain of the integrative analysis? What are exactly the main microbiota taxa and metabolites that were not found in any of the individual studies? For instance, gut microbiota depletion or reduced microbial diversity has been reported before in several publications. Amino acids have been associated before to IBD.

Response: Thank you for the comments. In the process of conducting cross-cohort integrative analysis, we take into account both gut microbiota and metabolite data simultaneously, resulting in the emergence of the following two novel insights.

(a) Construction and validation of a novel non-invasive diagnostic model for IBD using new fecal biomarkers across different cohorts.

Although previous studies has utilized fecal biomarkers for diagnosing IBD^{1,2}, there are still two unresolved issues: the reliable reproducibility of biomarkers obtained from fecal samples across different cohorts and populations, and whether it's possible to further enhance the diagnostic performance of the existing fecal diagnostic model.

In this study, we integrated metagenomic and metabolomic data from multiple cohorts to identify 31 species, 25 KO genes and 13 metabolites distinguishing normal control from IBD

cases (Fig 2F-G, Fig 3B-C and Fig 4G-H in revised manuscript). These biomarkers demonstrate robust reproducibility across various cohorts. Moreover, through the integration of diverse omics data, we have achieved a significant enhancement in the performance of our machine learning models for diagnosing IBD (Fig 6 in revised manuscript and Table R1, below).

Table R1: The AUROC values from different studies in IBD.

Study	Features	AUROC
Franzosa, E. A. et al. ¹	Species	0.90
	Metabolites	0.92
	Combined	0.92
Vich Vila, A. et al. ²	Metabolites	0.83
This study	Species (RJ cohort)	0.93
	Metabolites (RJ cohort)	0.94
	KO genes (RJ cohort)	0.90
	Combined (RJ cohort)	0.98

(b) New therapeutic targets for IBD

Through Cross-cohort Integrative Analysis (CCIA), we successfully identified three specific microbial species, *Asaccharobacter celatus*, *Gemmiger formicilis* and *Erysipelatoclostridium ramosum*, (Fig 2E and Extended Data Fig 2C, D in revised manuscript) linked to inflammation and immunity modulation, yet there is currently no literature reporting their association with IBD before.

Furthermore, we identified 162 differentially expressed KO genes between normal and IBD patients, followed by an enrichment analysis revealing 12 pathways potentially relevant to both gut microbiota and the disease (FDR < 0.05). Among the pathways, Two-component systems and Propanoate metabolism play a critical role in certain bacteria³⁻⁶, however, their association with IBD has not been reported yet (Extended Data Fig 3A-C in revised manuscript).

Additionally, compared to the external dataset, we have identified 36 newly discovered differential fecal metabolites within the internal dataset. Several metabolites are related to the consumption of red meat, such as 1-Methylhistidine and carnitine compounds. Some metabolites are associated with the tricarboxylic acid cycle, such as Fumaric acid, and others (Extended Data Fig 4D in revised manuscript). The roles of these metabolites in IBD are still unknown.

Moreover, by constructing Multi-Omics Biological Correlation (MOBC) maps of the gut microbiota in IBD, we revealed the characteristics of functional impairments in gut microbial biotransformation and significant variations in multiple Aminoacyl-tRNA synthetases (ARSs) within gut bacteria (Fig 5 in revised manuscript).

In conclusion, through our CCIA analysis of multi-cohorts and omics datasets, we have identified significant alterations in microbial species, functional genes of the gut microbiota, and fecal metabolites. These findings go beyond what could be detected in previous single-cohort or single-omics studies. However, additional investigation is needed to explore and confirm these findings.

Comment 2: The authors try to assess the disease-specific microbiome signature by analyzing data from three other diseases: GI, CRC and T2D. However, the LOCO classification models seem biased and needs further validation, including metabolites. The metabolomic signature seems very unspecific and could be associated to other diseases.

Response: Thank you for your constructive suggestions. To validate the disease specificity of our feature metabolites, we incorporated four additional metabolomics cohorts, including one adenoma cohort, two CRC cohorts, and one T1D cohort. Differential analysis revealed that the majority of the 13 metabolites, which were included for diagnosing IBD, did not show significant differences in these cohorts (**FDR>0.05, Extended Data Fig 4E-H in revised manuscript**). These findings substantiated the disease-specific nature of our featured metabolites in IBD.

Comment 3: The results from MOBC claiming abnormal ATP production by the microbiota in IBD patients is not supported by the data. Lines 374-384 are extremely speculative. There's no experimental evidence, neither from bacterial enzymes nor from the detected metabolites that could possibly indicate that the microbiota of IBM patients produces excessive amounts of ATP.

Response: Thank you for your helpful comments. We have revised the **Results section**. Please find the specific details in lines 408-421 of the revised manuscript.

Comment 4: The study focuses on 79 metabolites shared by the four cohorts (2 external cohorts by non-targeted metabolomics, and 2 in-house cohorts by using targeted metabolomics); however, it is unclear how metabolites were annotated/identified in the non-targeted datasets. Were they annotated based on MS1 data alone, or MS1 and MS2 data? MS2 data and matching scores should be provided as supplementary information. In general, the paper contains many vague interpretations in the results section.

Response: Thank you for your comment. The untargeted metabolomics data is derived from previously published sources, and this data was obtained from the supplementary materials of their papers, which did not describe whether the metabolomics data is annotated based on MS1 data or MS2 data¹. In mass spectrometry technology, when two consecutive mass analyzers perform secondary fragmentation, the ions before the secondary fragmentation are referred to as parent ions (MS1), and the ions formed after the secondary fragmentation are called daughter ions (MS2)⁷. Within the methodology section of the paper, the authors introduced the mass spectrometer they utilized as the Q Exactive Hybrid Quadrupole-Orbitrap Mass Spectrometer. This instrument combines the capabilities of both quadrupole and Orbitrap mass analyzers.

Therefore, we inferred that the authors employed MS2 annotations. However, the authors did not provide MS2 data and matching scores in the article; instead, they only presented a relative abundance table of metabolites derived from the final annotations. In this study, we directly utilized the data that had been pre-processed by the authors.

Comment 5: Minor: Lines 290-293: revise the text, the style is excessively baroque for scientific writing.

Response: Thank you for your helpful suggestion. We have revised this section in the updated manuscript. **Please find the specific details in lines 310-312 of the revised manuscript.**

Reviewer #2:

General Comments:

Comment 1: The article presents a comprehensive analysis of multiple cohorts to investigate the role of gut microbiota and metabolome in inflammatory bowel disease (IBD). The study aims to eliminate biases and confounding factors, such as race and diet, to provide valuable insights for future interventions and treatments based on microbiota or metabolites for IBD. Overall, the article presents interesting findings, it is highly relevant for the field of IBD, adding novel insight compared to available literature. It is well designed, sound, with a strong methodology which is one of the major innovation of this paper. However there are a few areas that could benefit from further clarification and discussion.

Strengths :

a. Large sample size: The inclusion of a substantial number of cases (1363 cases for metagenomic analysis and 398 cases for metabolomics analysis) enhances the reliability of the results.

b. Comprehensive approach: The integration of metagenomic and metabolomics analyses, as well as the construction of Multi-Omics Biological Correlation (MOBC) maps, provides a comprehensive understanding of the relationship between gut microbiota, microbial functional genes, and intestinal metabolites in IBD.

Response: Thank you for your comments.

Comment 2: Clarifications and Suggestions. please specify better the clinical characteristics of patient enrolled into the selected cohorts, such as demographic information (e.g., age, gender) and disease-related characteristics (e.g., disease subtype, Montreal Classification, disease duration) of the IBD patients. These informations are crucially relevant to better interpret the clinical meaning of the results.

Response: Thank you for your suggestion. We have added the clinical characteristics of patients in this study to **Supplementary Table 1** in revised manuscript.

Comment 3: Clarifications and Suggestions. Biases and limitations: Acknowledge and discuss potential limitations of the study. Address the limitations related to the cohort

selection process, sample collection, and analysis methods. Furthermore, discuss the potential impact of unmeasured confounders that might influence the observed associations.

Response: Thank you for your helpful suggestions. As the reviewer mentioned, being a cross-cohort study, it is challenging to completely avoid biases in aspects such as cohort selection, sample collection, and analysis methods. However, we have implemented the following approaches to minimize confounding factors to the greatest extent possible, ensuring the scientific rigor and validity of the study:

- (a) Cohort Selection Process: We ensured diverse and representative samples from 9 metagenomic cohorts and 4 metabolomic cohorts across Eastern and Western countries to mitigate bias.
- (b) Sample Collection: Stringent protocols maintained sample quality; participants provided ≥ 3.0 g stool, collected using sterile collectors (Thermo Scientific, USA, R21922). Samples were frozen at -80°C within 4 hours for further processing.
- (c) Analysis Methods: We used ANOVA-like analysis to assess confounding effects.

Of course, there might still be unmeasured confounding factors that could influence observed associations. For example, factors like dietary habits, medication usage, and lifestyle choices could concurrently impact both the gut microbiota and disease development, which need further validation in future research.

We have revised this section in lines 515-518 of the revised manuscript.

Comment 4: Interpretation of Findings. Mechanistic insights: Explain how future research could strengthen the findings of your study, postulating the kind of protocols and studies needed (i.e. longitudinal studies, interventional studies).

Response: Thank you for your comment. As the reviewer suggested, subsequent investigation can be structured as follows:

Firstly, analyze the ecological and biological characteristics of the microbial communities, genes, and metabolites, as well as their interactions with other microorganisms (fungi, viruses, etc.).

Subsequently, employ methods such as *in vitro* assays and animal models to conduct mechanistic studies, revealing their intricate interactions with the immune system and their impact on the progression of IBD.

Additionally, longitudinal studies could be considered to track the dynamic changes of these microbial communities, functional genes, and metabolites in IBD patients, aiding in a better understanding of their temporal relationship with disease evolution.

Finally, if feasible, implement intervention studies to assess the effects of manipulating these significantly altered components on disease development, further validating their therapeutic potential.

Comment 5: Interpretation of Findings. Diagnostic model: Provide more details regarding the development and validation of the diagnostic model.

Response: Thank you for your helpful suggestion. We have revised this section in the updated manuscript. Please find the specific details in lines 748-768 of the revised manuscript.

Comment 6: Discuss the potential clinical implications of the model and its performance in comparison to existing diagnostic methods.

The utilization of this multi-omics diagnostic model based on gut microbiota, functional genes, and metabolites in patients with IBD holds several clinical values. Firstly, it contributes to the early detection and prediction of IBD, facilitating timely interventions and reducing the risk of complications. Additionally, as a foundation for future research, this multi-omics model is poised to drive advancements in clinical practice and medical science. Most importantly, our findings hold potential as non-invasive diagnostic markers for IBD.

In contrast to the current invasive gold standard for diagnosing IBD, colonoscopic examination, we have demonstrated the potential of utilizing gut fecal microbiota and metabolites as a non-invasive approach to diagnosis. Compared to previous non-invasive diagnostic models^{1,2} (Table R1, see below), our model significantly enhances the performance of diagnosing IBD, achieving an AUROC value of 0.98.

Table R1: The AUROC values from different studies in IBD.

Study	Features	AUROC
Franzosa, E. A. et al. ¹	Species	0.90
	Metabolites	0.92
	Combined	0.92
Vich Vila, A. et al. ²	Metabolites	0.83
This study	Species (RJ cohort)	0.93
	Metabolites (RJ cohort)	0.94
	KO genes (RJ cohort)	0.90
	Combined (RJ cohort)	0.98

Comment 7: Future directions. Intervention and treatment strategies: Discuss how the identified alterations in gut microbiota and metabolome can guide the development of interventions and treatments for IBD. Are there any specific targets or pathways that appear particularly promising?

Response: Thank you for the suggestions. Through Cross-cohort Integrative Analysis (CCIA), we successfully identified three specific microbial species, *Asaccharobacter celatus*, *Gemmiger formicilis* and *Erysipelatoclostridium ramosum*, (Fig 2E and Extended Data Fig 2C, D in revised manuscript) linked to inflammation and immunity modulation, yet there is currently no literature reporting their association with IBD before.

Furthermore, we identified 162 differentially expressed KO genes between normal and IBD patients, followed by an enrichment analysis revealing 12 pathways potentially relevant to both

gut microbiota and the disease (FDR < 0.05). Among the pathways, Two-component systems and Propanoate metabolism play a critical role in certain bacteria³⁻⁶, however, their association with IBD has not been reported yet (**Extended Data Fig 3A-C in revised manuscript**).

Additionally, compared to the external dataset, we have identified 36 newly discovered differential fecal metabolites within the internal dataset. Several metabolites are related to the consumption of red meat, such as 1-Methylhistidine and Carnitines. Some metabolites are associated with the tricarboxylic acid cycle, such as Fumaric acid, and others (**Extended Data Fig 4D in revised manuscript**). The roles of these metabolites in IBD are still unknown.

Moreover, by constructing Multi-Omics Biological Correlation (MOBC) maps of the gut microbiota in IBD, we revealed the characteristics of functional impairments in gut microbial biotransformation and significant variations in multiple Aminoacyl-tRNA synthetases (ARSs) within gut bacteria (**Fig 5 in revised manuscript**).

In conclusion, through our CCIA analysis of multi- cohorts and omics datasets, we have identified significant alterations in microbial species, functional genes of the gut microbiota, and fecal metabolites. These findings go beyond what could be detected in previous single-cohort or single-omics studies. However, additional investigation is needed to explore and confirm these findings.

Comment 8: Consider the potential challenges and ethical considerations associated with interventions based on microbiota or metabolites.

Response: Thank you for your comment. Interventions based on microbiota or metabolites hold great promise for addressing various health conditions, but they also come with several potential challenges and ethical considerations that need careful consideration, such as microbiota variability, long-term effects and lack of standardization. **We have revised this section in lines 526-529 of the revised manuscript.**

Comment 9: Future directions. Longitudinal studies: Highlight the need for longitudinal studies to understand the dynamic changes in gut microbiota and metabolome during different stages of IBD and in response to interventions. This will provide insights into causality and the potential for personalized treatment approaches. In summary, the article provides valuable insights into the role of gut microbiota and metabolome in IBD. Addressing the suggested clarifications and discussing the interpretation of findings in relation to existing knowledge will further strengthen the article's impact and contribute to the field of IBD research.

Response: Thank you for your constructive suggestions. In the revised manuscript, we have emphasized the necessity for longitudinal studies to gain a deeper understanding of the dynamic changes in gut microbiota and metabolome during various stages of IBD and in response to intervention measures. **We have revised this section in lines 529-531 of the revised manuscript.**

Reviewer #3

Comment 1: The number for ethical approval should be given in the manuscript.

Response: Thank you for your comment. In the revised manuscript, we have provided the ethical approval number as requested. The patient cohorts were approved by the ethics committee of Renji Hospital affiliated to the School of Medicine, Shanghai Jiao Tong University, the ethical approval number is 2021-skt-004. We have revised this section in lines 828-832 of the revised manuscript.

Comment 2: The study describes enrolment of participants but is devoid of the basic reporting information associated with this. No information on numbers, timeframe enrolled, enrolment criteria are not adequately described. No description on how healthy controls were recruited.

Response: Thank you for your suggestion. We have revised the participant recruitment in the **Methods section**. Please find the specific details in lines 544-585 of the revised manuscript.

Comment 3: Details describing the samples collection are inadequate: “Stool samples were aseptically collected into fecal collection tubes and immediately preserved at -80°C until further processing” What were the collection tubes. What time were the samples collected – how long before being processed for storage. Details of medication use are not clear – were they taken into consideration?

Response: Thank you for your suggestion. We have added the protocol details for stool sample collection in the **Methods section of revised manuscript**. Please find the specific details in lines 598-603 of the revised manuscript.

Comment 4: The details on the recruitment of participants etc. do not have enough detail – the section on participant enrolment in the methods is devoid of detail. The registration of the study is not clear and the number of participants enrolled is not clear here or later in the paper. There are rigorous guidelines one should adhere to for the reporting of human participant data and this manuscript falls short. The reporting summary was not detailed enough.

Response: Thank you for your suggestion. We have made revisions in the **Methods section**. Additionally, in accordance with the requirements of the journal, clinical interventional cohort studies require registration, whereas our study is not belonging to this category and may not necessitate registration. Please find the specific details in lines 544-585 of the revised manuscript.

Comment 5: Figure 1- provides a nice summary of the cohorts – but does it warrant figure 1 in the paper – more appropriate in the supplementary data.

Response: Thank you for your suggestion. Considering that our study involves multiple omics data and employs complex machine learning model construction methods, we streamlined the workflow diagram in **Figure 1** of revised manuscript. The content of Figure 1 present the complex analytical process in a clearer and more intuitive way, making it easier to comprehend our analysis workflow and content.

Comment 6: The description of the metabolomic data is inadequate. For example metabolomic data is not sequenced yet the title is “metabolomic sequencing”

Response: Thank you for your suggestion. We have revised the manuscript. **Please find the specific details in lines 629 of the revised manuscript.**

Comment 7: Not enough information is given in terms of the identification of the metabolites and the level of identification. Methods says QC samples were used but no information in relation to the QC data is presented.

Response: Thank you for your comments. We have incorporated QC samples into the dataset in the revised manuscript.

Comment 8: The O-PLS-DA models for the metabolomic data needs to be validated – using permutation testing for example.

Response: We appreciate your suggestion regarding the validation of the O-PLS-DA models used for analyzing the metabolomic data. Through the permutation testing procedure, all the O-PLS-DA models are statistically significant (**Extended Data Fig 4A, C in revised manuscript**). **Please find the specific details in lines 723-725 of the revised manuscript.**

Comment 9: In general for the metabolomic data there is an over interpretation of the data – the data is faecal metabolite data which not only a representation of gut metabolites but instead has a complex mixture of metabolites originating from the host, gut microbes and ingested food. There is no appreciation of this in the manuscript and no attempt to differentiate metabolites that could potentially be microbe derived.

Response: Thank you for your insightful feedback on our manuscript. Indeed, the fecal metabolite data comprises a complex mixture of metabolites originating from the host, gut microbes, and ingested food. Therefore, we try to explore the metabolic processes in which the gut microbiota is involved. We not only deciphered functional genes related to gut microbial metabolism (KO genes) by filtering out host genes from the metagenome, but we have also constructed MOBC maps to unveil metabolites in which the gut microbes are involved. The variations in these genes suggest that the alternation of certain gut microbes is associated with these metabolic processes. This signifies our initial exploratory endeavor to distinguish metabolites that may potentially originate from microorganism.

Comment 10: The novelty of the current work is not evident. There are previous results showing alterations in amino acids and keto acids in IBD patients. While cohorts is important work the novelty is lower due to previous published work in this area.

Response: Thank you for the comments. In the process of conducting cross-cohort integrative analysis, we take into account both gut microbiota and metabolite data simultaneously, resulting in the emergence of the following two novel insights.

(a) Construction and validation of a novel non-invasive diagnostic model for IBD using new fecal biomarkers across different cohorts.

Although previous studies has utilized fecal biomarkers for diagnosing IBD^{1,2}, there are still two unresolved issues: the reliable reproducibility of biomarkers obtained from fecal samples across different cohorts and populations, and whether it's possible to further enhance the diagnostic performance of the existing fecal diagnostic model.

In this study, we integrated metagenomic and metabolomic data from multiple cohorts to identify 31 species, 25 KO genes and 13 metabolites distinguishing normal control from IBD cases (**Fig 2F-G, Fig 3B-C and Fig 4G-H in revised manuscript**). These biomarkers demonstrate robust reproducibility across various cohorts. Moreover, through the integration of diverse omics data, we have achieved a significant enhancement in the performance of our machine learning models for diagnosing IBD (**Fig 6 in revised manuscript and Table R1, below**).

Table R1: The AUROC values from different studies in IBD.

Study	Features	AUROC
Franzosa, E. A. et al.¹	Species	0.90
	Metabolites	0.92
	Combined	0.92
Vich Vila, A. et al.²	Metabolites	0.83
This study	Species (RJ cohort)	0.93
	Metabolites (RJ cohort)	0.94
	KO genes (RJ cohort)	0.90
	Combined (RJ cohort)	0.98

(b) New therapeutic targets for IBD

Through Cross-cohort Integrative Analysis (CCIA), we successfully identified three specific microbial species, *Asaccharobacter celatus*, *Gemmiger formicilis* and *Erysipelatoclostridium ramosum*, (**Fig 2E and Extended Data Fig 2C, D in revised manuscript**) linked to inflammation and immunity modulation, yet there is currently no literature reporting their association with IBD before.

Furthermore, we identified 162 differentially expressed KO genes between normal and IBD patients, followed by an enrichment analysis revealing 12 pathways potentially relevant to both gut microbiota and the disease (FDR < 0.05). Among the pathways, Two-component systems and Propanoate metabolism play a critical role in certain bacteria³⁻⁶, however, their association with IBD has not been reported yet (**Extended Data Fig 3A-C in revised manuscript**).

Additionally, compared to the external dataset, we have identified 36 newly discovered differential fecal metabolites within the internal dataset. Several metabolites are related to the consumption of red meat, such as 1-Methylhistidine and Carnitines, etc. Some metabolites are associated with the tricarboxylic acid cycle, such as Fumaric acid, and others (**Extended Data Fig 4D in revised manuscript**). The roles of these metabolites in IBD are still unknown.

Moreover, by constructing Multi-Omics Biological Correlation (MOBC) maps of the gut microbiota in IBD, we revealed the characteristics of functional impairments in gut microbial biotransformation and significant variations in multiple Aminoacyl-tRNA synthetases (ARSs) within gut bacteria (**Fig 5 in revised manuscript**).

In conclusion, through our CCIA analysis of multi-cohorts and omics datasets, we have identified significant alterations in microbial species, functional genes of the gut microbiota, and fecal metabolites. These findings go beyond what could be detected in previous single-cohort or single-omics studies. However, additional investigation is needed to explore and confirm these findings.

Comment 11: The results from the multiomic biological correlation over state that results – conclusions such as “The results from MOBC revealed functional impairments in gut microbial biotransformation and abnormal ATP production capacity in pathogenic bacteria” cannot be made without actual measurements of ATP or the functional capacity of the ATP production system. Changes in the TCA cycle intermediates cannot infer this alone.

Response: Thank you for your helpful comments. We have revised the **Results section**. Please find the specific details in lines 408-421 of the revised manuscript.

References

1. Franzosa, E. A. *et al.* Gut microbiome structure and metabolic activity in inflammatory bowel disease. *Nat Microbiol* **4**, 293–305 (2019).
2. Vich Vila, A. *et al.* Faecal metabolome and its determinants in inflammatory bowel disease. *Gut* gutjnl-2022-328048 (2023) doi:10.1136/gutjnl-2022-328048.

3. Duan, C. *et al.* Fucose promotes intestinal stem cell-mediated intestinal epithelial development through promoting Akkermansia-related propanoate metabolism. *Gut Microbes* **15**, 2233149 (2023).
4. van Hoek, M. L., Hoang, K. V. & Gunn, J. S. Two-Component Systems in Francisella Species. *Front Cell Infect Microbiol* **9**, 198 (2019).
5. Casado, J., Lanas, Á. & González, A. Two-component regulatory systems in Helicobacter pylori and Campylobacter jejuni: Attractive targets for novel antibacterial drugs. *Front Cell Infect Microbiol* **12**, 977944 (2022).
6. Sultan, M., Arya, R. & Kim, K. K. Roles of Two-Component Systems in Pseudomonas aeruginosa Virulence. *Int J Mol Sci* **22**, 12152 (2021).
7. Wang, X., Shen, S., Rasam, S. S. & Qu, J. MS1 ion current-based quantitative proteomics: A promising solution for reliable analysis of large biological cohorts. *Mass Spectrom Rev* **38**, 461–482 (2019).

REVIEWER COMMENTS

Reviewer #1 (Remarks to the Author):

My main points and concerns have been satisfactorily addressed.

Yet, I find surprising how one of the main results of the previous version claiming abnormal ATP production by the microbiota in IBD patients has completely disappeared in the revised manuscript.

Overall, I think the manuscript is now more robust and less speculative.

Reviewer #2 (Remarks to the Author):

Authors fulfilled to all requests.

Reviewer #3 (Remarks to the Author):

Its not clear where the info on the QC sample is added. Its now mentioned in the methods – but would expect some data presented in the results.

Information on identification level of metabolites is still lacking. Was MS2 used for the IDs. At the moment the information given in 641-643 is not sufficient for the in-house data.

Description of the data collected for this study is still lacking.

No info on the age criteria for recruitment and table indicates that some participants were <18.

It is still not clear how many were recruited and how they were then included for the different analyses. I think this whole section needs to be much more transparent. For example the description states that for the “a total of 208 participants were enrolled, comprising 138 patients diagnosed with Inflammatory Bowel Disease (IBD) and 70 healthy control subjects, matched for sex and gender. Specifically, the Puxi cohort (N=132, control=45, IBD=87) and the Pudong cohort (N=76, control=25, IBD=51) were employed for both model discovery and validation purposes.”

Yet the description for the PuDong cohort for the metabolomics states 52 IBD patients? This and other inconsistencies need to be addressed.

Why are there different numbers between the metagenomic and metabolomic datasets?

The ethical approval number is “2021-skt-004.” Could the authors confirm that ethical approval was in place when recruitment commenced in 2019 as stated in the methods section.

The statement “The P-value for each metabolite was corrected for FDR using the two-sided Mann-Whitney U-test.”. The Mann-Whitney test doesn't correct for FDR.

Dear reviewers,

We are very grateful for the positive and comprehensive review. The reviewer provided a large number of valuable and useful suggestions. In responses, we only focus on his/her concerns, and present a revised version of the manuscript. Our intention is to properly match the referees' comprehensive review of our paper with equally comprehensive responses, and address each of the referee's collective concerns, in detail, below. We directly include **Two figures** in the response letter. Our point-to-point responses are as follows:

Reviewer #1:

Major concerns:

Comment 1: My main points and concerns have been satisfactorily addressed. Yet, I find surprising how one of the main results of the previous version claiming abnormal ATP production by the microbiota in IBD patients has completely disappeared in the revised manuscript. Overall, I think the manuscript is now more robust and less speculative.

Response: Thank you for the comments. We excluded the result of this section from our study because two previous reviewers suggested that the results are not supported by our data. By reexamining our bioinformatics analysis and previous studies, we subsequently found a more robust conclusion regarding the significant involvement of aminoacyl-tRNA synthetases derived from the gut microbiota in IBD.

In addition, as Reviewer One suggested, we reanalyzed the metagenomic sequencing data. The abundance of two enzymes in bacterial ATP synthesis were enriched in IBD. The detailed results are presented as below, and we have no problem incorporating these data into revised manuscript upon request:

“Alongside the alterations in ARSs, we observed the involvement of Adenosine Triphosphate (ATP) in these reaction processes (**Fig 5G-J in revised manuscript**). This finding suggests that ATP may play a significant role in the development of IBD. Previous studies have reported that ATP not only functions as a universal energy currency¹ but also that extracellular ATP (eATP) activation of nucleotide receptors (purinergic P2 receptors) is a crucial pathway in the onset of inflammation². Specifically, gut commensal bacteria serve as a significant source of eATP^{3,4}, which has the potential to exacerbate progression of IBD⁵. Therefore, we investigated whether certain enzymes of ATP synthesis derived from commensal microbiota, are involved in IBD. We analyzed 17 enzymes which were involved in bacterial ATP synthesis. Among these enzymes, the abundance of K02113 (*atpH*, F-type H⁺-transporting ATPase subunit delta) and K02114 (*atpC*, F-type H⁺-transporting ATPase subunit epsilon) were significantly increased in multiple IBD cohorts (**Figure R1, see below**).”

Figure R1

Figure R1. Differential analysis of 17 enzymes involved in bacterial ATP synthesis across four IBD cohorts. The asterisks represented the statistical p-value (*P < 0.05; **P < 0.01; ***P < 0.001; ****P < 0.0001).

Reviewer #2:

Comment 1: Authors fulfilled to all requests.

Response: Thank you for your comments.

Reviewer #3

Comment 1: It's not clear where the info on the QC sample is added. Its now mentioned in the methods – but would expect some data presented in the results.

Response: Thank you for your comment. We have provided the QC sample in the result section (**Extended Data Fig 4A**) of revised manuscript.

Comment 2: Information on identification level of metabolites is still lacking. Was MS2 used for the IDs. At the moment the information given in 641-643 is not sufficient for the in-house data.

Response: Thank you for your suggestion. Targeted metabolomics profiling was conducted using the Q300 Metabolite Array Kit from Metabo-Profile Biotechnology of China, and detailed methods can be found in a previously published study (Xie, G. *et al*⁶). During the detection process, we employed both isotopic internal standards and reference standards to construct standard curves, thereby obtaining absolute concentrations of metabolites. Metabolite qualification was performed using specific ion pairs (simultaneously utilizing both parent ions and daughter ions) and retention times of reference standards. Specifically, these compounds belong to Level 1 of compound annotations based on the revised confidence levels established by the Compound Identification work group at the 2017 annual meeting of the Metabolomics Society⁷ (Brisbane, Australia). Please find the specific details in lines 629-649 of the revised manuscript.

Comment 3: Description of the data collected for this study is still lacking. No info on the age criteria for recruitment and table indicates that some participants were <18.

Response: Thank you for your suggestion. We have the detailed description of the age criteria for recruitment, which was set between 16 and 65 years, aiming to maintain consistency with external cohort population in the revised manuscript. Please find the specific details in lines 553-554 of the revised manuscript.

Comment 4: It is still not clear how many were recruited and how they were then included for the different analyses. I think this whole section needs to be much more transparent. For example the description states that for the “a total of 208 participants were enrolled, comprising 138 patients diagnosed with Inflammatory Bowel Disease (IBD) and 70 healthy control subjects, matched for sex and gender. Specifically, the Puxi cohort (N=132, control=45, IBD=87) and the Pudong cohort (N=76, control=25, IBD=51) were employed for both model discovery and validation purposes.” Yet the description for the PuDong cohort for the metabolomics states 52 IBD patients? This and other inconsistencies need to be addressed. Why are there different numbers between the metagenomic and metabolomic datasets?

Response: Thank you for your suggestion. According to the inclusion and exclusion criteria shown in the **Methods section**, a total of 215 participants were recruited for our cohorts. As a result of sample quality control procedures conducted prior to metagenomic sequencing, seven subjects were unable to undergo complete metagenomic analysis due to insufficient fecal DNA. Additionally, 37 subjects were unable to undergo metabolomic analysis due to insufficient

quantities of freeze-dried feces. Thus, there was a difference in the number of subjects between our metagenomic and metabolomic datasets. Please find the specific details in the revised manuscript (line 572-585) and **Figure R2** (see below).

Figure R2

Figure R2. Recruitment workflow for in-house IBD Renji cohorts.

Comment 5: The ethical approval number is “2021-skt-004.” Could the authors confirm that ethical approval was in place when recruitment commenced in 2019 as stated in the methods section.

Response: Thank you for your comment. We confirmed that this study obtained ethical approval from the Ethics Committee at Shanghai Jiao Tong University School of Medicine before conducting the recruitment. Our research team has been focusing on the study of IBD since 2012, and the Renji Cohort was granted approval for recruitment at that time. In 2019, we published our first article about IBD⁸. Subsequently, with the aim of further expanding the cohort, the Renji Cohort has been granted a renewed ethical approval number in 2021(2021-

skt-004). We have provided the earlier approval number of the Renji Cohort in revised manuscript. Please find the specific details in lines 840-844 of the revised manuscript.

Comment 6: The statement “The P-value for each metabolite was corrected for FDR using the two-sided Mann-Whitney U-test.”. The Mann-Whitney test doesn’t correct for FDR.

Response: Thank you for your suggestion. We have revised the manuscript. Please find the specific details in lines 723-725 of the revised manuscript.

References

1. Khakh, B. S. & Burnstock, G. The double life of ATP. *Sci Am* **301**, 84–90, 92 (2009).
2. Idzko, M., Ferrari, D. & Eltzschig, H. K. Nucleotide signalling during inflammation. *Nature* **509**, 310–317 (2014).
3. Daisley, B. A. *et al.* Emerging connections between gut microbiome bioenergetics and chronic metabolic diseases. *Cell Reports* **37**, 110087 (2021).
4. Inami, A., Kiyono, H. & Kurashima, Y. ATP as a Pathophysiologic Mediator of Bacteria-Host Crosstalk in the Gastrointestinal Tract. *Int J Mol Sci* **19**, 2371 (2018).
5. Scott, B. M. *et al.* Self-tunable engineered yeast probiotics for the treatment of inflammatory bowel disease. *Nat Med* **27**, 1212–1222 (2021).
6. Xie, G. *et al.* A Metabolite Array Technology for Precision Medicine. *Anal. Chem.* **93**, 5709–5717 (2021).
7. Blaženović, I., Kind, T., Ji, J. & Fiehn, O. Software Tools and Approaches for Compound Identification of LC-MS/MS Data in Metabolomics. *Metabolites* **8**, 31 (2018).
8. Ma, D. *et al.* CCAT1 lncRNA Promotes Inflammatory Bowel Disease Malignancy by Destroying Intestinal Barrier via Downregulating miR-185-3p. *Inflamm Bowel Dis* **25**, 862–874 (2019).